# A transcriptional activator effector of *Ustilago maydis* regulates hyperplasia in maize during pathogen-induced tumor formation

Weiliang Zuo ®[1] ✉, Jasper R. L. Depotter[1,4], Sara Christina Stolze ®[2], Hirofumi Nakagami ®[2,3] & Gunther Doehlemann ®[1] ✉

*Ustilago maydis* causes common smut in maize, which is characterized by tumor formation in aerial parts of maize. Tumors result from the de novo cell division of highly developed bundle sheath and subsequent cell enlargement. However, the molecular mechanisms underlying tumorigenesis are still largely unknown. Here, we characterize the *U. maydis* effector Sts2 (Small tumor on seedlings 2), which promotes the division of hyperplasia tumor cells. Upon infection, Sts2 is translocated into the maize cell nucleus, where it acts as a transcriptional activator, and the transactivation activity is crucial for its virulence function. Sts2 interacts with ZmNECAP1, a yet undescribed plant transcriptional activator, and it activates the expression of several leaf developmental regulators to potentiate tumor formation. On the contrary, fusion of a suppressive SRDX-motif to Sts2 causes dominant negative inhibition of tumor formation, underpinning the central role of Sts2 for tumorigenesis. Our results not only disclose the virulence mechanism of a tumorigenic effector, but also reveal the essential role of leaf developmental regulators in pathogen-induced tumor formation.

Plant pathogens secrete effectors to cross-talk with hosts for their benefits. To reshape the host transcriptome, some pathogens exploit effectors to directly manipulate host gene regulation in two main mechanisms. One is characterized by transcription activator-like (TAL) effectors, first described in the plant pathogenic *Xanthomonas sp*[1-3]. These effectors contain a nucleus localization signal, tandem repeat DNA binding domain and transcriptional activation domain, which can function independently as transcription factors to activate host gene expression. On the other hand, several effectors are not transcription factor themselves, but control host gene expression through interacting with host transcription factors[4], recruit the suppressors of

transcription factors[5-8], or disrupt the assembly of transcription units[9,10], eventually leading to the inhibition or activation of host gene expression.

*Ustilago maydis* is a fungal pathogen which causes common smut in maize. It infects all the aerial maize organs, grows locally, causes tumor formation and produces massive amount of teliospores[11,12]. Consequently, *U. maydis* manipulates plant cell proliferation and creates additional space to form tumors to reside in. On maize seedling leaves, tumors consist of hyperplasic tumor cells from the de novo division of highly differentiated bundle sheath tissue, some of such hyperplasic cells together with mesophyll enlarged to become

[1]Institute for Plant Sciences and Cluster of Excellence on Plant Sciences (CEPLAS), University of Cologne, Cologne 50674, Germany. [2]Protein Mass Spectrometry, Max-Planck Institute for Plant Breeding Research, Carl-von-Linné Weg 10, 50829 Cologne, Germany. [3]Basic Immune System of Plants, Max Planck Institute for Plant Breeding Research, Cologne 50829, Germany. [4]Present address: Bioinformatics and Biostatistics, The Francis Crick Institute, 1 Midland Road, London NW1 1AT, UK. ✉e-mail: wzuo@uni-koeln.de; g.doehlemann@uni-koeln.de

hypertrophic tumor cells[13]. The mechanisms of host tumor formation and the causative effectors are still largely unknown. Until now, the functionally characterized effectors in *U. maydis* are mainly involved in immunity inhibition[14–17] or host metabolism manipulation[18,19]. See1 is the only effector identified that is directly involved in tumorigenesis by re-activating DNA synthesis[20]; deletion of See1 causes the inhibition of *U. maydis* induced hyperplasia cell division[13]. Different tumor cells have a very different physiology and transcriptome in between, and compared to the sourced host cells[13,21]. One therefore can hypothesize that *U. maydis* possesses effectors to directly, or indirectly manipulate the host transcriptome. In line with this, several effectors of *U. maydis* have been identified to target Topless transcription co-repressors to modulate hormonal signaling and expression of immune genes during infection[6–8]. However, little is known about how effectors modulate the host transcriptome to induce tumors. A detailed *U. maydis* transcriptome analysis revealed the temporal regulation of effector genes throughout all steps of infection[22], and laser-captured microdissection of different tumor cells coupled with RNA-seq showed spatial, cell type specific regulation of effectors[13,23]. The cross-species analysis between *U. maydis* and *Sporisorium reilianum*, the closest pathogenic smut relative which also infects maize but does not cause tumors, disclosed the differential regulation of effector orthologs that contribute to the distinct pathogenic development in the two species. A CRISPR-Cas9 mediated effector ortholog knock-in experiment discovered a functional diversification of an effector orthogroup UMAG_05318 - sr10075/ sr10079 during speciation[24].

In this study, we functionally characterized the effector Sts2 (UMAG_05318, Small tumor on seedlings 2). A *U. maydis* knockout strain for Sts2 (CR-Sts2) initiates tumor formation, but the tumors fail to expand due to reduced cell division of the bundle sheath. We discovered that Sts2 is a transcriptional activator, being translocated into host cell nucleus to activate the expression of leaf developmental regulators, especially those involved in bundle sheath development. Sts2 interacts with ZmNECAP1, a yet uncharacterized maize protein, which interacts with the adapter protein complex (AP2) and also has a transcriptional activation function. Our findings disclose a tumorigenic mechanism where a small *U. maydis* effector functions as a transcriptional activator in the host nucleus to rewrite the host developmental process by hijacking the key leaf developmental regulators.

## Results

### Sts2 regulates hyperplastic tumor cell induction by *U. maydis*

We generated an open reading frame shift knockout of *UMAG_05318* in *U. maydis* strain SG200 by using CRISPR-Cas9 mutagenesis. The resulting mutant showed reduced virulence (Fig. 1a) as it was described previously for a gene deletion strain[23] (Supplementary Fig. 1a). Similarly, CRISPR-Cas9 mutagenesis of *UMAG_05318* in a compatible pair of *U. maydis* wild type strains (FB1 and FB2) resulted in reduced virulence when compared to the wild type (Supplementary Fig. 1b). Distinct from the typical bulged tumors caused by *U. maydis* SG200, tumors induced by the UMAG_05318 knockout mutant failed to expand in size (Fig. 1a).

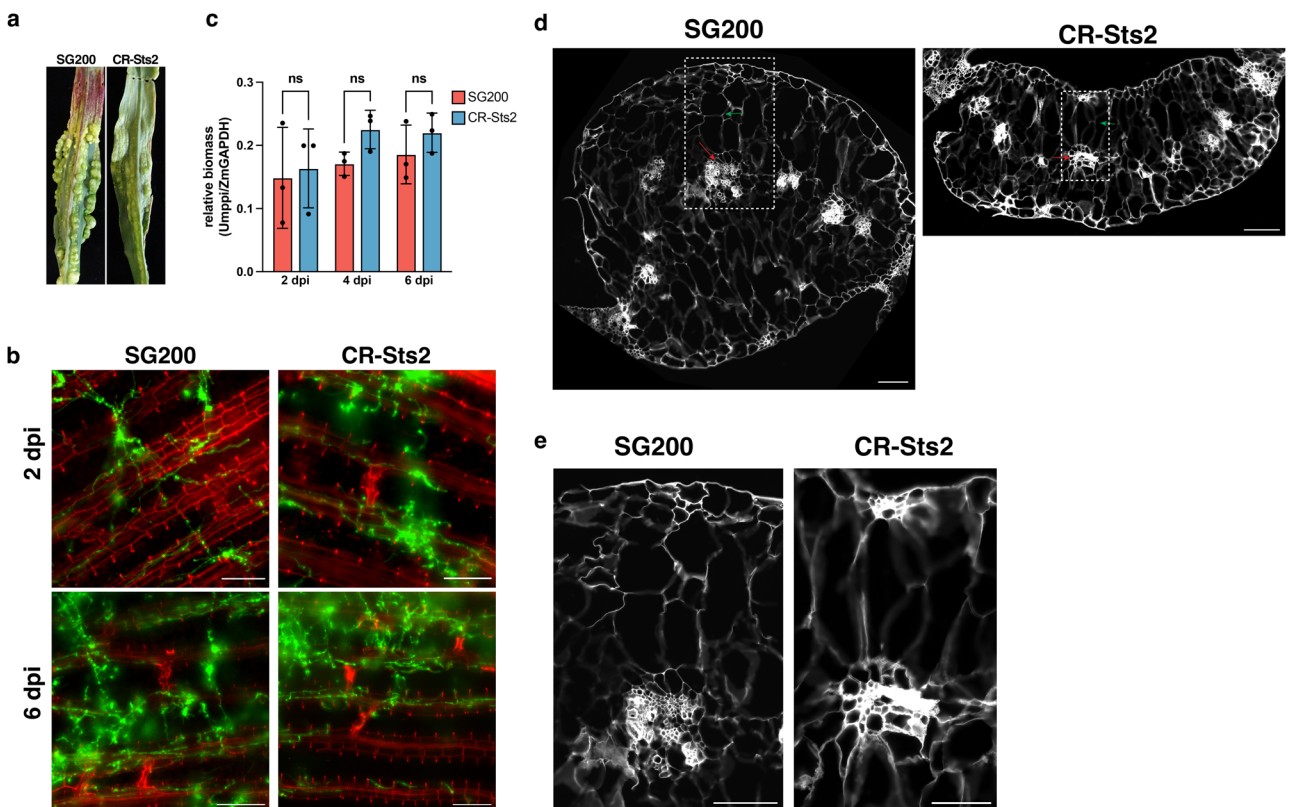

**Fig. 1 | Sts2 regulates the hyperplastic tumor cell formation. a** Photo shows the difference of the phenotype between SG200 and CR-Sts2 infected GB maize cultivar at 12 dpi, representative leaves were photographed. **b** Microscopic photos show the infected maize cells from SG200 and CR-Sts2 at 2 and 6 dpi. Green indicates hyphae from WGA-AF488 staining, and red is cell wall stained by propidium iodide. The experiments were repeated at least three times. Representative photos are shown. Scale bar = 100 μm **c** qPCR shows the relative biomass between SG200 and mutant. Data shown are the mean value ± SD from 3 biological replication of 3 independent infection. Two-tailed Student's t-test was used to determine the significance. ns, not significant. **d** Transverse leaf sections from 12 dpi illustrate the different tumor cells in SG200 and CR-Sts2 infected leaves between two large lateral veins. The autofluorescence of cell wall from DAPI channel was shown. The red arrows point out the hyperplasia tumor cells at the original bundle sheath position. The green arrows denote the hypertrophy tumor cells. The experiments were repeated for three times. Representative photos are shown. Scale bar = 200 μm. **e** Magnified views of vascular tissue from the dashed rectangle region in (**d**). Scale bar = 200 μm.

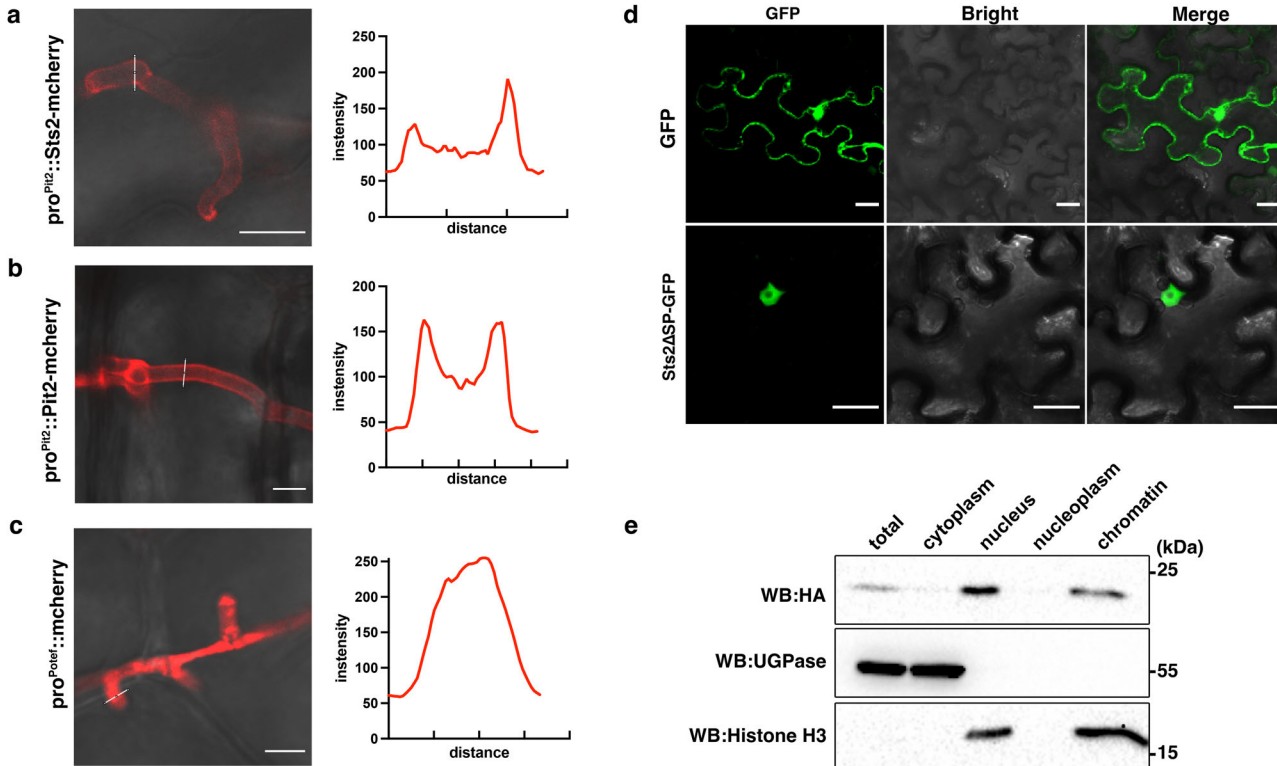

**Fig. 2 | Sts2 is secreted and translocated into host cell nucleus.** *U. maydis* hyphae expressing Sts2- mCherry (**a**), Pit2- mCherry (**b**) and mCherry (**c**) at 2 dpi. The plots are the mCherry intensities measured from the solid lines indicated in the photos. The experiments were repeated for two times. Representative photos are shown. Scale bar = 10 μm. **d** Localization of GFP and Sts2-GFP in *N. Benthamiana*. The experiments were repeated for two times Scale bar = 50 μm. **e** Western blot detection of Sts2-3 × HA in different cell fractions from 4 dpi maize leaves. UDP-Glucose-Pyrophosphorylase (UGPase) and histone H3 were used as the markers of cytoplasmic and nucleus/chromatin-associated fractions, respectively. The positions of molecular weight ladder are shown on the right. The experiments were repeated for three times with similar results.

Therefore, we named UMAG_05318 as Sts2 (S̲mall t̲umor on s̲eedlings 2). Expression of *Sts2* starts from 1 dpi (days post infection), and peaks between 2-4 dpi, when the tumor induction is initiated (Supplementary Fig. 1c)[22]. Wheatgerm agglutinin-Alexa Fluor 488 (WGA-AF488) and propidium iodide co-staining showed no obvious colonization defect in the CR-Sts2 mutant compared to SG200 at 2 and 6 dpi (Fig. 1b). We also monitored the growth of the mutant by relative fungal biomass quantification to 6 dpi (Fig. 1c), the timepoint when leaf tumors get visible. Here, we did not detect a growth difference between CR-Sts2 and SG200, suggesting that the defect of tumor formation is not caused by a compromised biotrophic growth of the mutant, i.e. that Sts2 is probably not an inhibitor of host immunity.

To further investigate how tumor formation is affected in the CR-Sts2 mutant, we embedded 12 dpi leaves for transverse sectioning. In SG200 infected samples, clusters of small hyperplasia tumor cells from de novo division of bundle sheath cells can be observed. These were surrounded by 5-6 layers of supernumerary hypertrophic cells from enlarged divided cells and mesophyll (Fig. 1d, e, Supplementary Fig. 1d), which is in accordance to our previous observation[13]. On the contrary, in CR-Sts2 infected plants, the de novo division was restricted, which resulted in depletion of hyperplasia cells, as well as reduced layers of hypertrophic cells (Fig. 1d, e, Supplementary Fig. 1d). In brief, knock-out of Sts2 causes a premature stop of bundle sheath cell division, which blocks further tumor expansion.

## Sts2 is secreted and translocated into the host nucleus

Sts2 encodes a small protein containing 183 amino acids (aa), including a 27 aa N-terminal signal peptide (SP). To test for secretion, we expressed Sts2-mCherry in the CR-Sts2 mutant under control of the *pit2* promoter, which confers a high expression level throughout plant infection (Fig. 2a)[25]. In confocal microscopy, Sts2-mCherry accumulated on the edge of biotrophic hyphae (similar to the effector control Pit2-mCherry[25]) indicating secretion during maize infection. (Fig. 2a, b). In contrast, the mCherry alone was localized in the cytoplasm of hyphae cell (Fig. 2c). Next, we determined the subcellular localization of Sts2 in the plant by transient expression of Sts2^ΔSP-GFP in *Nicotiana benthamiana* (Fig. 2d). The specific nuclear localization of Sts2^ΔSP suggests that Sts2 might be translocated into the host cells upon secretion by *U. maydis* (Fig. 2d). To test this, we complemented the *U. maydis* CR-Sts2 mutant with a Pro^Sts2::Sts2-3×HA construct. The expression of the Sts2-3xHA fusion protein fully restored virulence, indicating that the C-terminal HA-tags did not affect the virulence function of Sts2. 4 dpi leaves from Pro^Sts2::Sts2-3×HA infection were collected and followed up with cell fractionation to detect subcellular localization of Sts2-3×HA (Fig. 2e). By western blot, we could detect Sts2 in the nuclear component, more precisely in the fraction associated with chromatin pellet after nucleoplasm extraction (Fig. 2e). Together, this data shows that Sts2 is a secreted *U. maydis* effector which is translocated into the host nucleus.

## Sts2 has *trans* activator function

To elucidate the molecular function of Sts2, we attempted to use the yeast two hybrid system to identify potential maize interactors. When we fused Sts2 with Gal4 DNA binding domain (BD), the recombinant protein exhibited a strong activation of reporter genes and grew on dropout plate compared to empty vector control or BD-See1^ΔSP (Fig. 3a), which implied that Sts2 may have a transactivation function. Using the 9aaTAD tool (https://www.med.muni.cz/9aaTAD/), we identified two overlapping nine aa (DTATANAAL and ATANAALQP) transactivation domains (TAD) in Sts2 (Fig. 3b). To test the function of

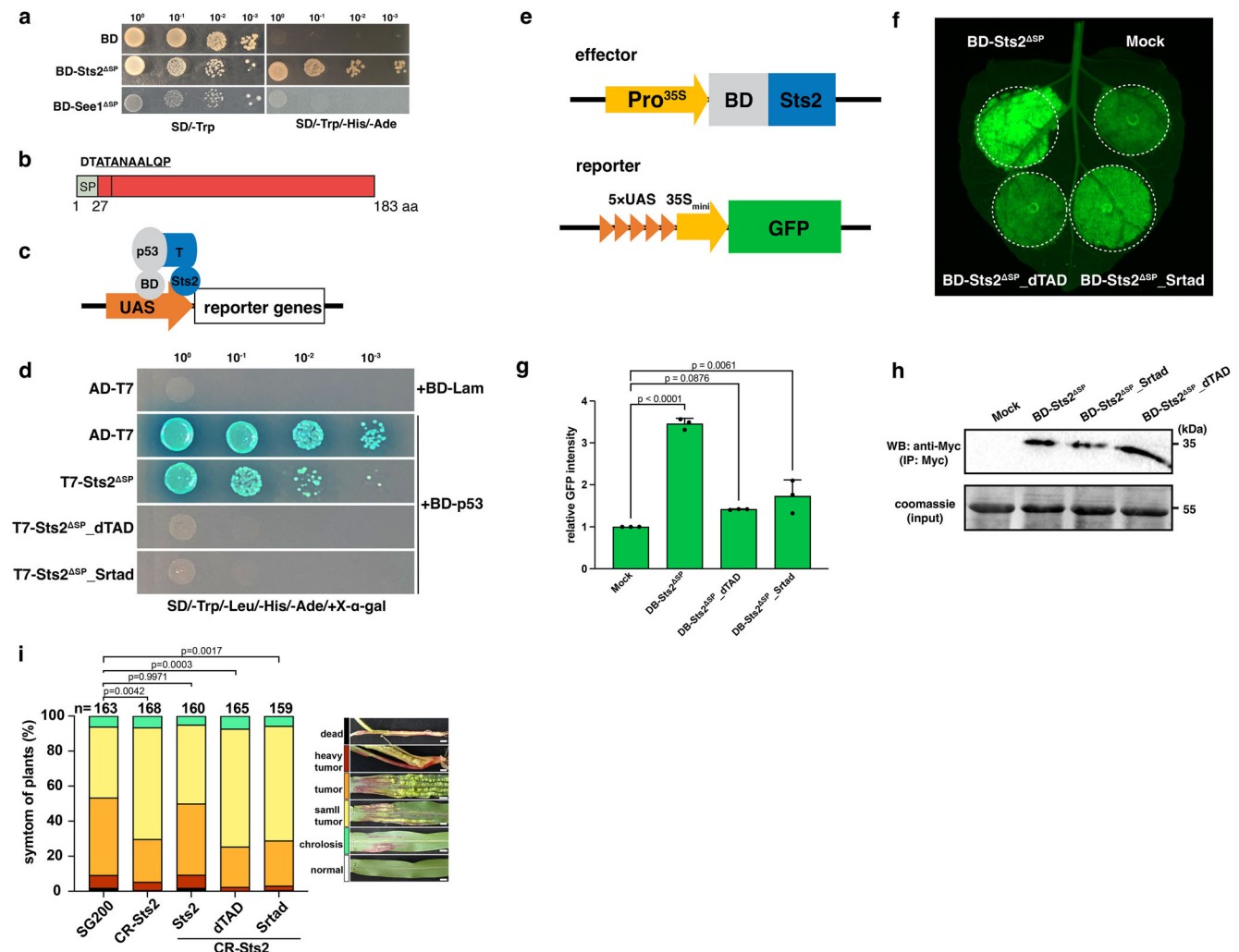

**Fig. 3 | Sts2 is a transcriptional activator. a** The photo shows the BD-Sts2$^{\Delta SP}$ transformed yeast grown on dropout SD medium plate. Strains transformed with empty vector (BD) and BD-See1$^{\Delta SP}$ are shown as a control. **b** Domain arrangement of Sts2. The vertical line shows the position of overlapping TADs, and the amino acid sequences are shown above. The underlined sequences indicate the domain mutated or deleted in the following experiments. **c** Diagram explains the co-transformation of BD-p53 and T7- Sts2$^{\Delta SP}$ and the interaction between p53 and T7 antigen brings Sts2$^{\Delta SP}$ to the promoter of reporter genes to activate their expressions. **d** The photo shows the growth of yeasts transformed with corresponding constructs on dropout SD plate. **e** Diagram shows the construct used for agroin-filtration. **f** GFP detection in *N. benthamiana* leaf co-infiltrated with the corre-sponding constructs together with Pro$^{5\times UAS\text{-}35Smini}$::GFP. The dashed circles indicate the infiltration area. **g** Relative GFP intensity from three biological replications. The intensity was normalized to the mock. Data shown are the mean value ± SD from 3 biological replication of 3 independent infiltrations. Dunnett's 1-way ANOVA test was used for significance analysis. **h** Western blot shows the protein expression levels of BD- Sts2$^{\Delta SP}$ and its mutants in the infiltrated leaf. The BD- Sts2$^{\Delta SP}$ protein and its mutants were enriched by the anti-Myc beads before detection. 5% of total extracts were loaded as input. The experiments were repeated for three times with similar results. **i** The disease symptoms of SG200, CR-Sts2 and complementation strains with Sts2 and different Sts2 variants. n is the total number of plants infected from 4 independent infections. Dunnett's 1-way ANOVA test was used for sig-nificance analysis based on the disease index data. The photos represent typical disease symptom of each category. Scar bar = 1 cm.

the predicted TAD, we mutated the domain ATANAALQP, since it was identified by "less stringent" and "pattern for clusters" models. We generated two different mutants by changing ATANAALQP into the corresponding aligned amino acids (EQAREHIQA) of the orthologous protein Sr10075 (Srtad) from *S. reilianum*, or deleting the entire motif (dTAD). To minimize the noise expression of the reported genes in yeast, we adapted the positive interaction control p53 and T antigen used in yeast two hybrid assay by fusing Sts2 with T antigen to replace the Gal4 activation domain (Fig. 3c). The interaction between p53 and antigen T brings Sts2 to the proximal of the promoter upstream the reporter genes thus activating their expression (Fig. 3d). As expected, co-transformation of BD-p35 with T-Sts2$^{\Delta SP}$_Srtad or T-Sts2$^{\Delta SP}$_dTAD failed to grow on dropout plate, indicating the loss of transactivation activity in the mutant proteins (Fig. 3d). In a next step, we tested the *trans* activation of Sts2 *in planta*. To this end, we set-up an effector-GFP

reporter system in *Nicotiana benthamiana* (Fig. 3e). In consistency with the yeast experiments, BD-Sts2$^{\Delta SP}$ significantly induced GFP expression in *N. benthamiana* driven by the Pro$^{5\times UAS\text{-}35Smini}$ (Fig. 3f). This induction was significantly reduced or completely abolished, when the TAD was mutated or deleted (Fig. 3g, h). Thus, Sts2 is a transcriptional activator *in-planta*, and this activity depends on its TAD motif. A cru-cial question is, if this activity is necessary for the virulence function of Sts2. To test this, we genetically complemented *U. maydis* CR-Sts2 with either, wild-type Sts2, Sts2_Srtad or Sts2_dTAD mutant under its native promoter and used the resulting strains for plant infection assays. While wild-type Sts2 complementation completely restored virulence, strains expressing TAD-mutated versions showed reduced tumor for-mation similar to the CR-Sts2 (Fig. 3i). This suggests that the *trans* activation function of Sts2, mediated by the TAD, is crucial for its virulence function.

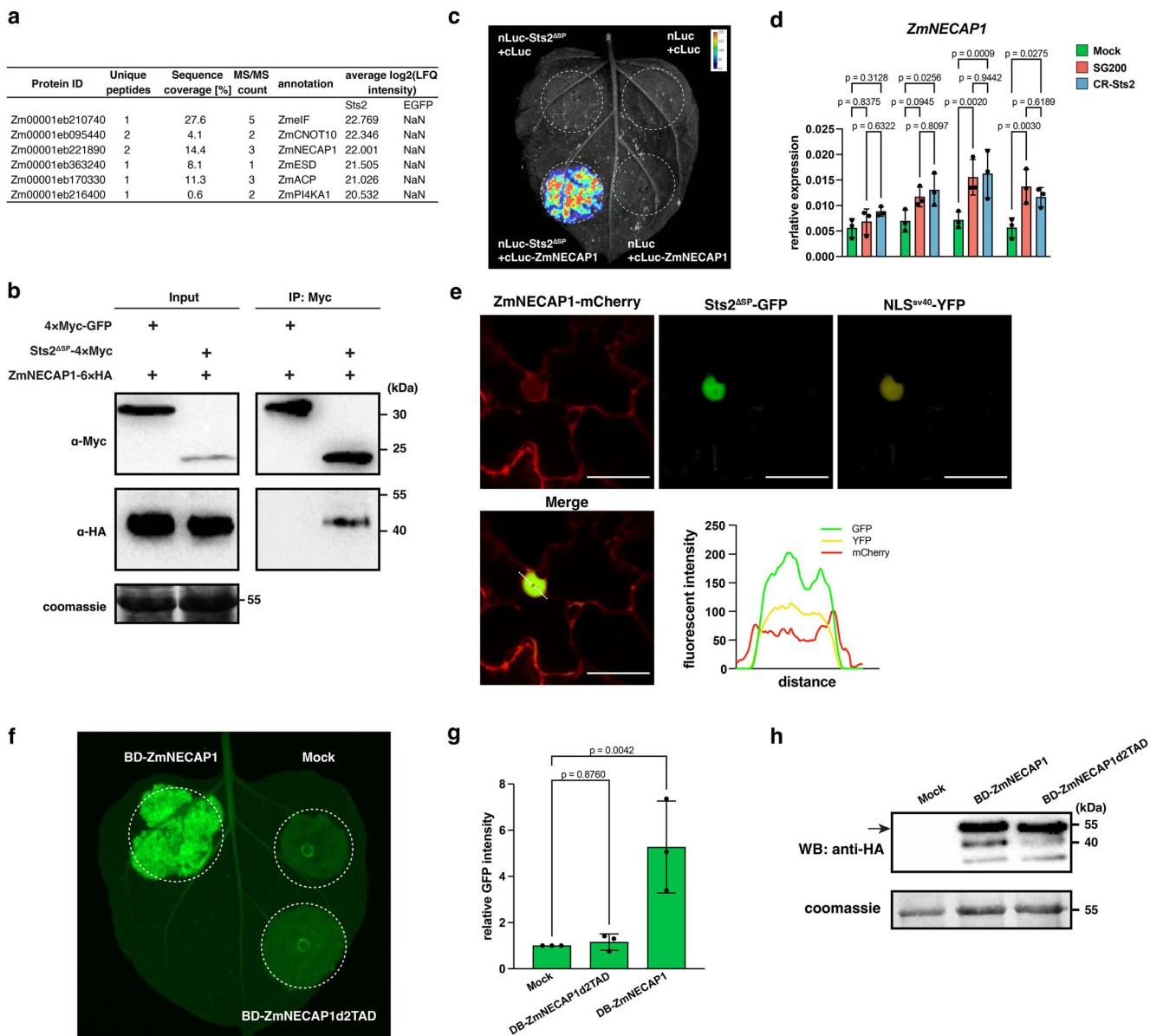

**Fig. 4 | Sts2 interact with a maize transcription activator ZmNECAP1. a** The list shows the average log2(LFQ intensity) of 6 candidates in 4 biological replications of Sts2-3HA samples and SP-EGFP-3HA controls. ZmeIF, translation elongation/initiation factor. ZmCNOT10, CCR4-NOT transcription complex subunit 10. ZmNECAP1, adaptin-ear-binding coat-associated protein 1. ZmESD, esterase D. ZmACP, acyl carrier protein. ZmPI4KA1, Phosphatidylinositol 4-kinase alpha 1. NaN, not a number, indicates that the protein was not detected in the samples in the MS. **b** Western blot of co-immunoprecipitation experiment in *N. Benthamiana* shows the interaction of Sts2 and ZmNECAP1.Commasie image shows around 1.38% of input. The experiments were repeated for three times with similar result. **c** Split luciferase complementary assay in *N. Benthamiana* shows the interaction between Sts2$^{\Delta SP}$ and ZmNECAP1. The circles indicate the infiltration area. **d** qPCR reveals the *ZmNECAP1* expression during *U. maydis* SG200 and CR-Sts2 infection. Data shown are the mean value ± SD from 3 biological replication of 3 independent infection. Tukey Two-Way ANOVA was used for significance test. **e** Microscope photos show the co-localization of Sts2$^{\Delta SP}$ and ZmNECAP1 and NLS$^{SV40}$-YFP in nucleus. The red is ZmNECAP1-mCherry, yellow is NLS$^{SV40}$-YFP and the green is Sts2$^{\Delta SP}$-GFP. The GFP, YFP and mCherry intensities were measured from the solid white line as shown in the "Merge". The experiments were repeated for three times. Representative photos are shown. Scale bar = 20 μm. **f** The photo shows the expression of BD-ZmNECAP1 activated Pro$^{5\times UAS-35Smini}$::GFP. The dashed circles indicate the infiltration area. **g** The bar plot shows the GFP intensity from three biological replications of three independent infiltrations. Data shown are the mean value ± SD and normalized to the Mock. Dunnett's 1-way ANOVA test was used to determine the significance. **h** Western blot shows the protein expression level of BD-ZmNECAP1 and BD-ZmNECAP1d2TAD from infiltrated leaves. Black arrow indicates the band with the expected size. The experiments were repeated for three times with similar result.

## Sts2 interacts with ZmNECAP1, a putative maize transcriptional activator

Sts2 does neither contain canonical known DNA binding domains, nor nuclear localization signals, which suggests that it is a transcriptional activator, and participates in the hosts gene regulation network. To further explore how Sts2 activates host gene expression, we performed a co-immunoprecipitation (co-IP) to identify the interacting host proteins. For this, CR-Sts2 expressing the Sts2-3×HA was infected

to maize leaves for subsequent sample preparation. As a control, *U. maydis* SG200 expressing GFP-3×HA (with the N-terminal secretion signal of Sts2, and expressed under control of the native sts2 promoter) was used. The immunoprecipitated proteins were subjected to mass spectrometry for protein identification. In total, 6 maize proteins were exclusively detected in Sts2-3×HA samples but not in the GFP-3×HA controls from at least 3 independent biological replicates (Fig. 4a, Supplementary Data 1). Of these, we could independently

confirm the interaction of Sts2 with Zm00001eb221890 both by co-IP (Fig. 4b) and split luciferase complementation assay in *N. benthamiana* (Fig. 4c). Zm00001eb221890, which is predicted as an adaptin-ear-binding coat-associated protein 1 (ZmNECAP1 hereafter) containing a pleckstrin homology domain, shows expression in maize leaves, and displays significant transcriptional induction upon *U. maydis* infection (Fig. 4d, Supplementary Fig. 2a), however, the expression levels were not changed between SG200 and CR-Sts2 infection (Fig. 4d). Subcellular localization identifies ZmNECAP1 in the cytoplasm, nucleus and nuclear membrane (Fig. 4e, Supplementary Fig. 2b), while in the nucleus ZmNECAP1 and Sts2 were co-localized (Fig. 4e).

To our surprise, ZmNECAP1 contains two separated TADs (Supplementary Fig. 2c). Similar to Sts2, a BD-ZmNECAP1 activated the reporter GFP expression under Pro$^{5\times UAS-35Smini}$ in *N. benthamiana* (Fig. 4f–h), while deletion of the TADs completely abolished this activity (Fig. 4f–h). Thus, not only Sts2, but also its maize interactor ZmNECAP1 is a transcriptional activator. To investigate whether the interaction affects the transactivation activity of Sts2, we co-transformed BD-Sts2$^{\Delta SP}$ with NLS$^{Gal4}$-ZmNECAP1 into yeast strain AH109 and measured the activity of reporter gene β-galactosidase. This resulted in a small, yet significant increase of β-galactosidase activity in BD-Sts2$^{\Delta SP}$/NLS$^{Gal4}$-ZmNECAP1 compared to BD-Sts2/EV, suggesting that the interaction of Sts2 with ZmNECAP1 may enhance its transactivation activity (Supplementary Fig. 2d).

In mouse, NECAP1 is enriched in clathrin-coated vesicles[26] and interacts with the α and β subunit of adapter complex protein AP2 via its WxxF and PH-like domain, respectively[27]. Moreover, recent findings showed that effector-uptake into plant cells involves clathrin-mediated endocytosis[28,29]. We therefore tested for an interaction between ZmNECAP1 and maize AP2 adapter β subunit via co-IP. Indeed, this experiment confirmed that ZmNECAP1 can interact with maize AP2β (Supplementary Fig. 2e).

## Sts2 activates maize leaf developmental regulators for tumor formation

To determine the host genes potentially regulated by Sts2, we conducted RNA-sequencing. For this purpose, *U. maydis* SG200, CR-Sts2 infected and mock-treated leaves from 3 and 6 dpi were sampled and analyzed. The infection of *U. maydis* dramatically altered maize leaf transcriptome. More than 45.2% of genes (11394 and 13593 genes at 3 and 6 dpi, respectively) were differentially expressed between mock treated and SG200 infected samples (Supplementary Fig. 3a). In general, CR-Sts2 triggered a similar maize transcriptional change at different timepoints to that by SG200 (Supplementary Fig. 3a), which confirms that mutation of Sts2 did neither affect the biotrophic growth of the mutant, nor trigger increased maize immune responses. In total, 5035 genes and 2370 genes were up-regulated or down-regulated, respectively, during the *U. maydis* infection, regardless of the genotype and time point (Supplementary Figure. 3a, b). Gene ontology (GO) enrichment analysis identified genes involved in several cell cycles related biological processes were both up-regulated by SG200 and CR-Sts2 infection (Supplemental Data 2), which is in line with that the de novo cell division of bundle sheath were initiated, but prematurely stopped by knockout as shown by transverse section microscopic photo.

To further elucidate key factors that control the sustained hyperplasia cell division, we compared the maize DEG between SG200 and CR-Sts2 infection samples in pairwise. In total, 435 and 465 genes were significantly up-regulated in SG200 samples at 3 and 6 dpi respectively, compared to 271 and 339 genes in CR-Sts2 samples (Fig. 5a, b). GO enrichment analysis revealed that at 3 dpi, genes involved in the "stem cell population maintenance" and "meristem maintenance" are specifically activated by *U. maydis* infection in presence of Sts2 (Fig. 5c). At 6 dpi, in addition to "meristem maintenance", several developmental processes were up-regulated depending on

Sts2 (Fig. 5c). Interestingly, from the 83 genes that were consistently, significantly up-regulated in SG200 infected samples at both time points (Fig. 5b), we identified the expression level of several transcription factors and activators were intensified in the presence of Sts2, including *ZmGRF3* (growth-regulating factor 3), *ZmGIF1* (growth-regulating-factor-interacting factor 1), *ZmYAB1* (YABBY1), *ZmSHR1* (short root 1), *ZmWOX5b* (WUSCHEL-Homeobox-transcription factor 5b) and *ZmANT1* (AINTEGUMENTA 1) (Fig. 5d). These genes are highly expressed in the dividing zone during leaf development[30] and in maize embryonic leaf cells[31,32]. ZmGRF interacts with ZmGIF, together to determine the shoot meristem in maize[33–35], where ZmANT1, ZmSHR1 and ZmYAB1 are regulators of Kranz anatomy development[31,32,36] and WOX members are well known highly expressing in embryo like cells[37]. Overexpression of some of these genes alone can lead to excess cell division in Arabidopsis[38], rice[39] and Medicago[40]. While these genes are highly induced upon SG200 infection compared to mock samples, knock out of Sts2 resulted in a strongly reduced expression (Supplementary. 4). Furthermore, CR-Sts2 strains complemented with mutated TAD (Srtad and dTAD) also failed to activate the expression of these genes to the levels observed for SG200. Similarly, *U. maydis* strains where UmSts2 was replaced by the open reading frames of the two *S. reilianum* Sts2 orthologs (KI_sr10075 and KI_sr10079) could not induce these maize genes during infection. (Fig. 5e). Together, this shows that the activation of these maize genes requires Sts2 with its functional transactivation motif.

To test, if the reduced expression of these genes is specific to Sts2 regulation rather than being a consequence of compromised tumor formation, we checked the expression of these Sts2-induced genes in other *U. maydis* effector mutants being compromised in tumorigenesis. Infection of maize inbred line CML322 with an *U. maydis* deletion mutant for the effector gene *UMAG_02297* resulted in reduced tumors[41]. Re-analysis of corresponding RNA-seq data reveals that *ZmGIF1, ZmGRF3, ZmYAB1, ZmSHR1*, and *ZmANT1* were not significantly changed in the *UMAG_02297* knockout compared to SG200, although expression of *ZmWOX5b* could not be detected (Supplementary Fig. 5a). More importantly, qPCR showed that induction of the Sts2-induced maize genes is not affected upon *U. maydis* ΔSee1 mutant infection (Supplementary. 5b), which displays a similar arrested hyperplasia tumor division phenotype to CR-Sts2[13,20]. All together, these findings confirm a specific induction of the identified maize genes by Sts2.

Following our conclusion that Sts2 induces the expression of maize genes required for tumorigenesis, we hypothesized that the transcriptional repression of Sts2-regulated maize genes would result in an inhibition of tumor formation. However, as the maize mutants of these genes are not available, we decided for the alternative approach to fuse a transcriptional suppressor SRDX motif with Sts2. The SRDX motif was shown to turn transcriptional activators into suppressors and moreover, it could inhibit the *trans* activation of its interacting transcription factors /activators[42,43]. Thus, an Sts2-SRDX fusion is expected to act as a dominant suppressive effector. Indeed, in the effector-GFP reporter system, BD-ZmNECAP1 activated the expression of GFP under the control of Pro$^{5\times UAS-35Smini}$::GFP as shown above (Fig. 4d, e). However, upon co-infiltration with Sts2$^{\Delta SP}$-SRDX, this activation was suppressed by the interaction between ZmNECP1 and Sts2$^{\Delta SP}$-SRDX as expected, but not by the Sts2 fused with mutated SRDX motif (SRDXm) (Fig. 5f). To test the effect of Sts2-SRDX mediated repression in the actual *U. maydis* - maize interaction, we generated *U. maydis* strains over-expressing Sts2-SRDX (or Sts2-SRDXm) to outcompete the native Sts2 and infected them to maize seedlings to evaluate the tumor formation. At 12 dpi, *U. maydis* strains expressing Sts2-SRDX, but not Sts2-SRDXm, showed significantly reduced the tumor formation. This reduction seemed to be dose dependent, as multiple integration of the Sts2-SRDX overexpression construct had an additive effect, resulting in further decreased tumor formation (Fig. 5g). qPCR showed that the

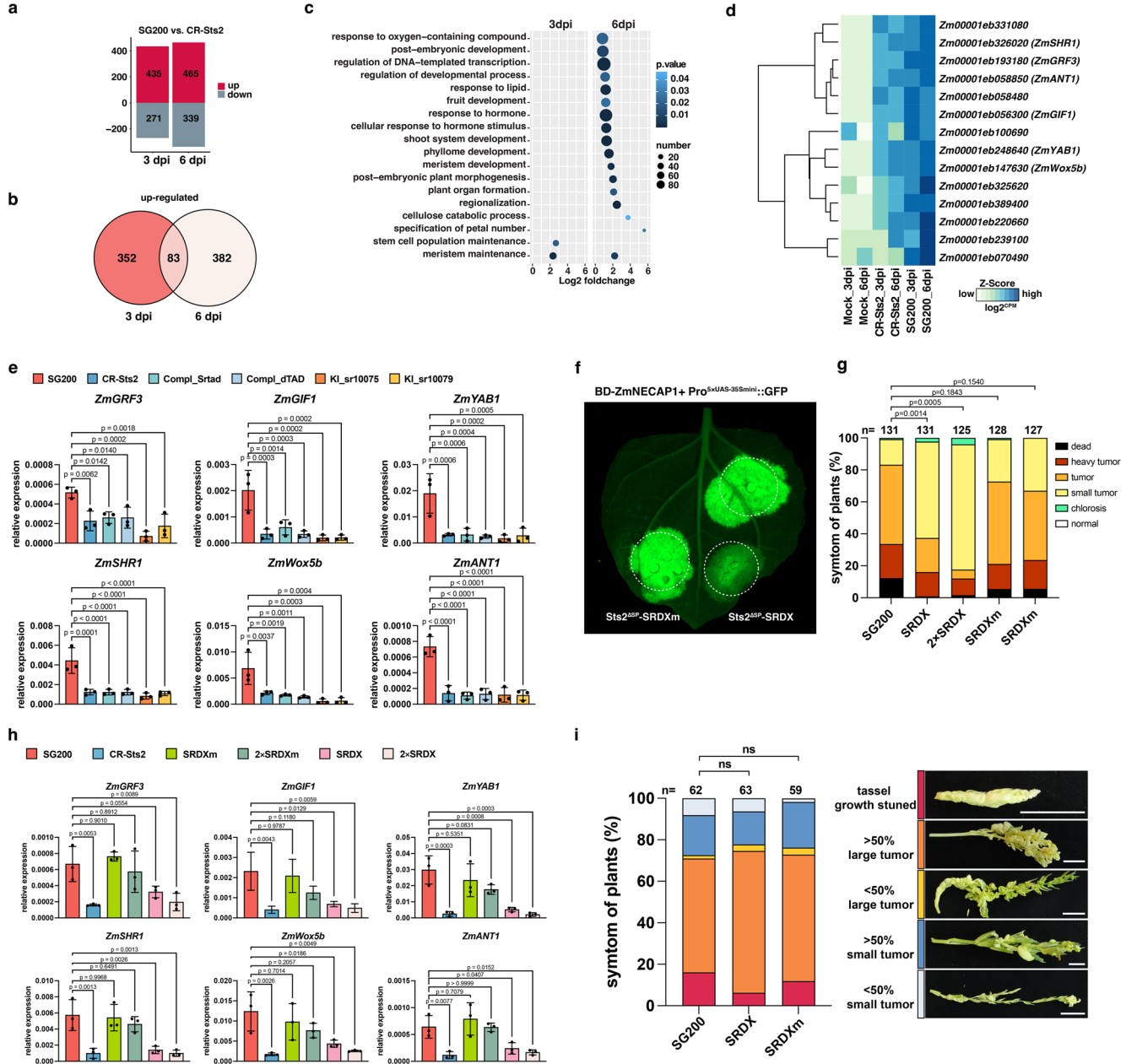

**Fig. 5 | Sts2 activates maize leaf development regulators during infection. a** Bar plot shows number of genes significantly up- or down-regulated in SG200 compared to CR-Sts2 at 3 and 6 dpi. **b** Venn diagram shows a total of 83 genes were consistently up-regulated upon SG200 infection from both timepoints compared to CR-Sts2. **c** The dot plot shows GO enrichment analysis of DEGs. **d** Heatmap shows the expression levels of differentially expressed transcription regulators at both timepoints between SG200 and CR-Sts2 infection. The Z score normalization was performed on the $\log_2$CPM (counts per million) of each gene across samples. **e** The quantitative RT-PCR results show the expression levels of *ZmGRF3*, *ZmGIF1*, *ZmYAB1*, *ZmSHR1*, *ZmWox5b* and *ZmANT1* upon infection of different strains. Data shown are the mean value ± SD from 3 biological replications of 3 independent infections, the solid points indicate the value of each replication. Dunnett's 1-way ANOVA test was used for significance analysis of the gene expression level between SG200 and different strains. **f** Sts2$^{\Delta SP}$-SRDX suppresses the transactivation function of ZmNECAP1. A level2 MoClo construct containing the Pro$^{5\times UAS-35Smini}$::GFP and BD-ZmNECAP1 was co-infiltrated with respective construct shown in the photo. The

dashed circles indicate the infiltration area. **g** The disease scoring of GB infected by SG200, and SG200 overexpressing Sts2-SRDX and Sts2-SRDXm. The experiment was repeated three times with independent infection, n is the total number of infected plants. Dunnett's 1-way ANOVA test was used for significance analysis based on the disease index data. **h** qPCR quantification of the expression levels of leaf developmental regulators upon SG200, SG200 overexpression strains Sts2-SRDX and Sts2-SRDXm infection at 6 dpi. Data shown are the mean value ± SD from 3 biological replications of 3 independent infections, the solid points indicate the value of each replication. Dunnett's 1-way ANOVA test was used for significance analysis of the gene expression level between SG200 and different strains infection. **i** Disease scoring of SG200, SG200 overexpression strains Sts2-SRDX, Sts2-SRDXm on tassel infection. The experiment was repeated in three independent infections; n is the total number of plants from 3 independent infections. Dunnett's 1-way ANOVA test was used for significance analysis based on the disease index data. ns, not significant. The represent photos of each disease symptom are shown in the legend. Scar bar = 1 cm.

transcript levels of Sts2-dependent leaf developmental regulators were significantly inhibited by Sts2-SRDX. On the contrary, Sts2-SRDXm did not significantly affect expression of the maize genes, independent of its copy number (Fig. 5h). Notably, overexpression of Sts2-SRDX or

Sts2-SRDXm did not completely suppress or further increase the expression of these genes (Fig. 5h).

We found previously that Sts2 contributes to fungal virulence during leaf infection, but not on tassel, where *U. maydis*-induced

tumor formation results from redirecting the intrinsic cell proliferation without an oncogenic activity[23]. In line with this, tassel infection of the Sts2-SRDX and Sts2-SRDXm mutants showed no reduction in virulence compared to strain SG200 (Fig. 5i). This underpins that the restricted tumor formation caused by Sts2-SRDX is not consequence of a general inhibition of *U. maydis* virulence, but results from the suppression of tissue specific host developmental regulators. Taken together, we conclude that the Sts2 effector is translocated from fungal hyphae to plant nuclei, where it contributes to tumor formation by activating the expression of leaf development regulators. In spite of that, heterologous overexpression of Sts2 in *S. reilianum* under a strong promoter (Pro[sr12761]) could not induce tumor formation during seedling infection (Supplementary Fig. 6a).

### Sts2 orthologs form a transcriptional activator family in the *Ustilaginales*

In a previous study, we found that *Sporisorium reilianum f. sp. zeae* contains two Sts2 orthologs (Sr10075 and Sr10079), which were not up-regulated during seedling leaf infection, and were functionally divergent from Sts2[24]. We found two different mechanisms underlying this function divergence, respectively. While Sr10079 completely lost its transcriptional activity (Supplementary Fig 6b, c), Sr10075 is a functional transcriptional activator and activated the expression of Pro[5×UAS-35Smini]::GFP to the level as Sts2 (Supplementary Fig 6b, c). However, Sr10075 did not activate the expression of leaf developmental regulators when expressed in *U. maydis* (Fig. 5e), suggesting that Sr10075 might regulate different genes during *S. reilianum* infection. BLAST search in the NCBI database identified Sts2 orthologs exclusively in *Ustilaginales* species, including *Ustilago trichophora*, *Kalmanozyma brasiliensis*, *Pseudozyma hubeiensis*, *Sporisorium scitamineum*, *Sporisorium. reilianum f. sp. reilianum* and *Sporisorium graminicola* (Supplementary Fig 6e, f). These species contain at least one Sts2 ortholog, either with, or without a paralog lacking a predicted TAD.The TAD containing Sts2 orthologs are more closely related, which suggests an early duplication before speciation. (Supplementary Fig 6e, f).

## Discussion

*U. maydis* induced tumor formation is a complex and still poorly understood process. During this process, *U. maydis* secretes a group of the tumorigenic effectors which not only prime highly differentiated bundle sheath cells for de novo cell division, but also sustains such division to increase the tumor size. Until now, See1 is the only effector which has been shown to re-activate the DNA synthesis, a prerequisite of cell division[20]. Surprisingly, *S. reilianum* SrSee1 protein resembles this function in *U. maydis*, but not UhSee1 from the barley smut fungi *Ustilago hordei*[20,44]. Here, we disclose Sts2 as another tumorigenic effector and, to our best knowledge, the first functionally validated fungal effector which acts as a transcriptional activator in the host. Moreover, we found that the virulence function of Sts2 depends on its transcriptional activation function. Compared to TALE effectors of bacterial pathogen *Xanthomonas sp*[3], Sts2 lacks a known repetitive DNA binding domain, which may be related to the generally small size of fungal effectors to facilitate the secretion into host cell via conventional secretory pathway. The lack of a known DNA binding domain also implies that Sts2 needs to recruit host components to form a transcriptional complex to activate target gene expression.

We found that Sts2 interacts with ZmNECAP1, a yet uncharacterized maize protein with transcriptional activator function. We have not yet fully understood, which role this protein has for the biological activity of Sts2. Based on current evidence, we speculate that Sts2 is a more efficient transactivator compared to ZmNECAP1, as a much lower Sts2 expression level of BD-Sts2[ΔSP] in *N. benthamiana* resulted in a comparable GFP level. Also, ZmNECAP1 might recruit and/or enhance the transactivator activity of Sts2, as the interaction with ZmNECAP1 enhances the transcriptional activation level of BD-Sts2 in yeast. On the

other hand, we found that ZmNECAP1 interacts with maize AP2 β subunit, a key adapter for clathrin-coated vesicles. Recently, it was revealed that clathrin-mediated endocytosis plays an important role in effector uptake[28,29]. Future investigations on a potential dual function of ZmNECAP1 during *U. maydis*-induced tumor formation and leaf development will aim to elucidate whether it is a co-activator with Sts2, or rather a mediator required for Sts2 uptake and/or to target DNA binding proteins.

*U. maydis* infection comprehensively reprograms the leaf developmental process to form tumors, which makes it hard to pinpoint the key host factors related to tumorigenesis. In this study, we show that Sts2 amplifies the expression of a group of plant transcription factors and activators regulating leaf development, after the developed bundle sheath cells are primed for division. This also partially explains why heterologous expression of Sts2 alone in *S. reilianum* did not induce hyperplasia tumor formation, since the effector(s) required to potentiate cell division of the developed bundle sheath cell is/are missing. We hypothesize that the Sts2-regulated genes are the potential executors to maintain hyperplasia tumor propagation. Accordingly, Sts2-SRDX specifically suppresses these genes, leading to compromised tumor formation on maize leaves, but not on tassel. It will be intriguing to know whether these genes are directly regulated by Sts2, or whether their regulations are decided downstream of Sts2 by yet unknown transcription factors. So far, we identified Sts2 orthologs only in a few smut fungi, either individually or with a paralog without predicted TAD. It will be interesting to test whether these orthologs are functional transcriptional activators and whether they are functionally conserved with Sts2. This will provide further insight into the mechanism of how smut fungi use transcriptional activator like effector to specifically manipulate the host genes. The results presented in this study demonstrate the potential of microbial effectors as molecular tools to help us explore complex cellular processes in host organisms, such as the network of co-regulation of leaf development regulators. Above all, we continue to be fascinated by the power of evolution to produce the molecular diversity of pathogen-host interactions necessary to adjust highly complex processes such as tumor formation in a cell type-specific manner.

## Methods
### Strains and plant material, growth conditions
All mutants were generated in *U. maydis* solopathogenic strain SG200 in this study (otherwise indicated). The *U. maydis* strains were grown on PD-agar plate at 28 °C or YEPS light liquid medium at 28 °C, 200 rpm. The maize cultivar Golden Bantam was used for infection (otherwise indicated). The plants were grown at controlled greenhouse or phytochamber with 16 hr light at 28 °C and 8 hr dark at 22 °C.

The CRISPR mutagenesis of Sts2 was done as previous described[24] by the oligo listed in Supplementary Data 3. The resulting strains were sent for sequencing to confirm the open reading frame shift and premature stop. For Sts2-SRDX or SRDXm, the amino acid sequence of SRDX (LDLDLELRLGFA)[43] and SRDXm (FDPDQEARFGFA)[42] were first codon optimized in the Eurofins online tool, and then assembled with Sts2 open reading frame and Pit2 promoter by Gibson assembly. The constructs were then introduced in SG200 by *ip* integration and further analyzed by southern blot and qPCR to check the integration and copy number (data not shown). The complementary of all CR-Sts2 with Sts2 and Sts2 mutants were done in a similar method.

### Transactivation activity test
For the autoactivation test in yeast, the yeast strain AH109 was used. The Sts2[28-183] was amplified and cloned into pGBKT7 to fuse with Gal4 BD domain, or into pGADT7-T to replace Gal4 AD domain and in open reading frame with antigen T protein. The resulting plasmids were transformed into AH109 alone or with pGBKT7-p53, respectively. The transformation and drop assay were done according to the protocol

from manufacturer. To test the β-galactosidase activity, ZmNECAP1 was cloned into Acc65I and BamHI digested pGADT-7 plasmid. The resulted plasmid was co-transformed with pGBKT7-BD-Sts2$^{\Delta SP}$. The β-galactosidase activity was measured according to the protocol from the manufacturer (Yeast Protocols Handbook, Clontech), and three independent colonies from each transformation were tested.

For the autoactivation test in *Nicotiana Benthamiana*, the Sts2$_{28-183}$, Gal4-BD domain and promoter 5×UAS-35Smini were cloned into MoClo system with BsaI and BpiI domestication[45], and further used for modular cloning of the corresponding constructs. The BD-Sts2$^{\Delta SP}$ and BD-ZmNECAP1 and their mutants were tagged with 4×Myc and 6×HA, respectively, for western blot detection. The constructs were transformed into agrobacterium strain GV3101 by electroporation. The resulting agrobacterium was grown overnight until OD$_{600}$ between 1-2, and suspended in infiltration buffer (10 mM MgCl$_2$, 10 mM MES pH5.6, 200 μM acetosyringone) to OD$_{600}$ = 3, incubated in dark for 2 hr. Before infiltration, an equal volume of each strain was mixed with p19 strain to final OD = 1. Leaves from 3 dai (days after infiltration) were used for GFP detection by ChecmiDoc MP machine (BioRad), and ImageJ Fiji was used for quantification of GFP intensity. The GFP intensity was normalized to the mock infiltration. Six 0.4 cm diameter leaf disks were collected from each infiltrated area for protein extraction and western blot.

### Maize infection and disease scoring

For seedling infection, seven-days-old Early Golden Bantam (EGB) seedlings or six-days-old Golden Bantam (GB) seedlings were used. The infection and disease scoring were done as previously described[24]. The disease index was used for statistic test, and Tukey multiple comparison test with Bonferroni adjustment was used for significance test. For tassel infection and scoring were done on maze cultivar Gaspe Flint according to Redkar et al.[46]. For significance test, a similar disease index was used as in seedling infection by assigning index 9 to tassel growth stunted, >50% large tumor (7), >50% small tumor (5), <50% large tumor (3), <50% small tumor and normal tassel as 0. Then the disease index was used for statistic test, and Dunnett's one-Way ANOVA test was used for significance test. The biological replications were from independent infection experiments.

### Leaf staining, tissue embedding, sectioning and microscope

Wheat germ agglutinin-Alexa Fluor 488 (WGA-AF488, Invitrogen, W11261) and propidium iodide (Invitrogen, P1304MP) co-staining was done according to previous description. For tissue embedding, around 1.8-cm leaf sections 3 cm below the infection sites were collected and embedded according to Alexandra M, 2017[13]. Leaf tissues were arranged in Peel-A-Way™ Einbett-Formen mold, and sectioned into 15 μm for microscope. The microscopy of staining tissues and transverse sections were done by using a Nikon Eclipse Ti inverted microscope with the Nikon Instruments NIS-ELEMENTS software. The GFP, mCherry and YFP microscopy was done by using a Leica TCS SP8 confocal laser scanning microscope.

### Co-immunoprecipitation and mass spectrometry in maize

Inoculums with OD$_{600}$ = 3 and 0.1% tween-20 were used to infect EGB. At 3 dpi, 4 cm length of leaf sections 1 cm below the infection site were collected and ground into fine powder with liquid nitrogen. The powders were incubated in extraction buffer (50 mM Tris-HCl pH7.5, 150 mM NaCl, 10% glycerol, 2 mM EDTA, 5 mM DTT, 1% Triton X-100 and protease inhibitor) for 30 min on ice and centrifuged twice at 16,000 g, 4 °C for 30 min. 10 μl of anti-HA magnetic beads (Pierce) were added into each supernatant and followed by 1 hr incubation at 4 °C with end-to-end rotation. Afterward, the beads were washed three times with extraction buffer and three times with extraction buffer without Triton X-100. In total, four replications from 4 independent infections were prepared and to MS analysis.

For MS analysis, dry beads were re-dissolved in 25 μL digestion buffer 1 (50 mM Tris, pH 7.5, 2 M urea, 1 mM DTT, 5 ng/μL trypsin) and incubated for 30 min at 30 °C in a Thermomixer with 400 rpm. Next, beads were pelleted and the supernatant was transferred to a fresh tube. Digestion buffer 2 (50 mM Tris, pH 7.5, 2 M urea, 5 mM CAA) was added to the beads, after mixing the beads were pelleted, the supernatant was collected and combined with the previous one. The combined supernatants were then incubated o/n at 32 °C in a Thermomixer with 400 rpm; samples were protected from light during incubation. The digestion was stopped by adding 1 μL TFA and samples were desalted with C18 Empore disk membranes according to the StageTip protocol[47].

Dried peptides were re-dissolved in 2% ACN, 0.1% TFA (10 μL) for analysis and diluted 1:10 for measurement. Samples were analyzed using an EASY-nLC 1000 (Thermo Fisher) coupled to a Q Exactive mass spectrometer (Thermo Fisher). Peptides were separated on 16 cm frit-less silica emitters (New Objective, 75 μm inner diameter), packed in-house with reversed-phase ReproSil-Pur C18 AQ 1.9 μm resin (Dr. Maisch). Peptides were loaded on the column and eluted for 115 min using a segmented linear gradient of 5% to 95% solvent B (0 min: 5%B; 0-5 min -> 5%B; 5-65 min -> 20%B; 65-90 min ->35%B; 90-100 min -> 55%; 100-105 min ->95%, 105-115 min ->95%) (solvent A 0% ACN, 0.1% FA; solvent B 80% ACN, 0.1%FA) at a flow rate of 300 nL/min. Mass spectra were acquired in data-dependent acquisition mode with a TOP15 method. MS spectra were acquired in the Orbitrap analyzer with a mass range of 300–1750 m/z at a resolution of 70,000 FWHM and a target value of $3\times10^6$ ions. Precursors were selected with an isolation window of 1.3 m/z. HCD fragmentation was performed at a normalized collision energy of 25. MS/MS spectra were acquired with a target value of $10^5$ ions at a resolution of 17,500 FWHM, a maximum injection time (max.) of 55 ms and a fixed first mass of m/z 100. Peptides with a charge of +1, greater than 6, or with unassigned charge state were excluded from fragmentation for MS$^2$, dynamic exclusion for 30 s prevented repeated selection of precursors.

### Co-immunoprecipitation in *N. Benthamiana*, subcellular fractionation and western blot

For co-IP experiments in *N. benthamiana*, domesticated *Bsa*I and *Bpi*I recognition site free Sts2$^{\Delta SP}$, ZmNECAP1 and ZmAP2β were cloned into MoClo level-0 plasmid pAGM1287 by using the primers listed in Supplementary Data 2. The resulting plasmids were further assembled with corresponding tags into level-1 binary vector and transformed into agrobacterium strain GV3101. The agroinfiltration and the co-IP was done as described above. Leaf samples from 2 dai were collected and ground to fine powder. The protein was extract by the following buffer (50 mM Tris-HCl pH7.5, 150 mM NaCl, 10% glycerol, 10 mM ETDA, 1 mM DTT, 1 mM PMSF, 1% IGEPAL CA-630 and protease inhibitor). The Myc-Trap magnetic agarose beads (ChromoTek) were used for immunoprecipitation.

The subcellular fractionation was done according to the method from Haring M, 2007[48]. Afterwards, the resulting nucleus pellet was split into two halves. One half was used for nuclei lysis followed as Chang L, 2018[49] with some modification. The nuclei pellet was suspended in 100 μl glycerol buffer (20 mM Tris-HCl, pH 7.9, 50% glycerol, 75 mM NaCl, 0.5 mM EDTA, 1 mM DTT, 1 mM PMSF and Roche protease inhibitor cocktail), then 100 μl prechilled nuclei lysis buffer (10 mM HEPES, pH 7.6, 1 mM DTT, 7.5 mM MgCl$_2$, 0.2 mM EDTA, 0.3 M NaCl, 1 M Urea, 1% IGEPAL CA-630, 1 mM PMSF, and Roche protease inhibitor cocktail) was added, vortexed and incubated on ice for 5 min. After centrifugation at 4 °C with maximum speed, the supernatant was taken as nucleoplasm fraction and the pellet was washed twice with cold PBS buffer containing 1 mM EDTA. The anti-Myc (Sigma, M4439) and anti-HA (Sigma, H-9658) antibody were used for detection by using ChecmiDoc MP machine (BioRad).

## Split Luciferase complementary assay

Different agrobacterium strains were mix with p19 to the final $OD_{600} = 1$, the leaves from *Nicotiana Benthamiana* were shortly rinsed in water, sprayed with 1 mM D-luciferin (Promega) and kept in dark for 10 min before detection by using ChecmiDoc MP machine (BioRad).

## DNA and RNA preparation and quantitative PCR

For RNA-seq, at least 15 leaves from individual plants were mixed as one sample, for biomass and gene expression quantification, one sample was mixed from at least 5 individual leaves. Each experiment was repeated three times from three independent infections. DNA was prepared by Buffer A (0.1 M Tris- HCl, 0.05 M EDTA, 0.5 M NaCl, 1.5% SDS) and purified by MasterPure Complete DNA and RNA Purification Kit Bulk Reagents (Epicenter, Madison, WI, USA). RNA was prepared by TRizol according to the manufacturer's protocol, followed by DNaseI digestion. The cDNA synthesis was done by using RevertAid First Strand cDNA Synthesis Kit (Thermo Scientific). The qPCR was done by using GoTaq qPCR mix (Promega) and performed on CFX96 Real-Time PCR Detection System (Bio-Rad). $2^{-\Delta Ct}$ ($Ct^{UmPpi}$-$CT^{ZmGAPDH}$) and $2^{-\Delta Ct}$ ($Ct^{GOI}$-$CT^{reference}$) was used to determine the relative biomass and gene expression, respectively and Student's *t-test* or Dunnett's one-Way ANOVA test was used for statistical analysis of significance.

## MS and RNA-seq data analysis

For MS data analysis, the raw data were processed using MaxQuant software (version 1.6.3.4, http://www.maxquant.org/)[50] with label-free quantification (LFQ) and iBAQ enabled[51]. MS/MS spectra were searched by the Andromeda search engine against a combined database containing the sequences from *Z. mays* (Zmays_284_Ensembl-18_2010-01-MaizeSequence.protein_primaryTranscriptOnly.fasta), the bait protein and sequences of 248 common contaminant proteins and decoy sequences. Trypsin specificity was required and a maximum of two missed cleavages allowed. Minimal peptide length was set to seven amino acids. Carbamidomethylation of cysteine residues was set as fixed, oxidation of methionine and protein N-terminal acetylation as variable modifications. Peptide-spectrum-matches and proteins were retained if they were below a false discovery rate of 1%. Statistical analysis of the MaxLFQ values was carried out using Perseus (version 1.5.8.5, http://www.maxquant.org/). Quantified proteins were filtered for reverse hits and hits "identified by site" and MaxLFQ values were log2 transformed. After grouping samples by condition only those proteins were retained for the subsequent analysis that had two valid values in one of the conditions. Two-sample *t*-tests were performed using a permutation-based FDR of 5%. The Perseus output was exported and further processed using Excel.

The RNA-seq was done in Novogene. Reads were filtered using the Trinity software (v2.9.1) option trimmomatic under the standard settings[52] and then mapped to the reference genome using Bowtie 2 (v2.3.5.1) with the first 15 nucleotides on the 5′-end of the reads being trimmed[53]. The reference genome was the genome assembly of *U. maydis*[54] combined with the assembly of *Z. mays* B73 version 5[55]. Reads were counted using the R package Rsubread (v1.34.7)[56]. The edgeR package v.3.26.8 was used for statistical analysis and determine the differential gene expression by using the pairwise comparison generalized linear models (GLMs). Genes with log2 fold change>1 or <−1 and $p < 0.05$ were considered as differentially regulated between SG200 and ΔSts2 samples. The differentially regulated genes between treatment were subjected to gene ontology analysis by using PLAZA 5.0 with default setting.

## Statistical analysis

The statistical analysis was done using GraphPad Prism 9 software.

## Reporting summary

Further information on research design is available in the Nature Portfolio Reporting Summary linked to this article.

## Data availability

Source data are provided with this paper. All data that support the findings of this study which are not directly available within the paper (and its supplementary information files) will be available from the corresponding authors (GD, WZ) upon reasonable request. RNAseq raw data are publicly accessible in the NCBI Gene Expression Omnibus with accession number GSE225929. The mass spectrometry proteomics data have been deposited to the ProteomeXchange Consortium via the PRIDE [ref- PMID: PXD040350] partner repository. Source data are provided with this paper.

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

## Acknowledgements

This project has received funding from the European Research Council (ERC) under the European Union's Horizon 2020 research and innovation program (grant agreement No 771035), as well as funding by the Deutsche Forschungsgemeinschaft (DFG, German Research Foundation) under Germany´s Excellence Strategy- EXC-2048/1- Project ID: 390686111 and Research Grant DFG-Az: DO 1421/3-3.

## Author contributions

W.Z. and G.D. designed the research; W.Z. performed all molecular experiment, virulence assay and RNA-seq data analysis; J.R.L.D. mapped the RNA-seq data; S.C.S. and H.N. conduced MS and MS data analysis. W.Z. and G.D. wrote the paper with contributions from other authors.

## Funding

## Competing interests

The authors declare no competing interests.
