## [Peer Review File · Nature Communications]

REVIEWER COMMENTS

Reviewer #1 (Remarks to the Author):

This manuscript by Zuo et al. presents that the *Ustilago maydis* effector Sts2 may act as a transcriptional activator to upregulate the expression of genes involved in maize leaf development, which leads to leaf tumor formation. As how *U. maydis*, a devastating maize pathogen, causes tumor formation is unknown, this study can have a broad significance.

The authors employ a multitude of approaches, such as cytohistology, comparative transcriptomics, transcriptional activity assay, and trans-repression assay, to reveal the functional mechanism of Sts2. Firstly, histological results indicated that Sts2 possibly promotes hyperplasia of bundle sheath cells in the leaf tissue without immune suppression. Secondly, they discovered that Sts2 is localized to the plant nucleus and exhibits transcriptional activator activity only when fused to a DNA-binding domain. A putative transcription factor of maize, ZmNECAP1, was identified as an interaction partner of Sts2. Lastly, the authors showed that the transcriptional activation activity of Sts2 is required to induce the expression of several maize genes involved in leaf development, which may explain leaf tumor formation.

Overall, this study is intriguing and reveals for the first time the transcriptional activator activity of *U. maydis* effectors, which extends our knowledge of the virulence mechanism of this model smut fungus. The characterized effector Sts2 is distinct from known *U. maydis* effectors as it exploits its transcriptional activator activity to upregulate regulators of maize leaf development for tumor formation without inhibiting plant immunity. The manuscript is well-written, and the data shown are generally solid. Nevertheless, the work has weakness in its current form, especially in the mechanistic aspect. Below are some comments for the authors to improve their manuscript:

Major comments:

1. The authors stated that Sts2 possibly interacts with other proteins to execute its activator activity as it does not contain a predicted DNA binding domain. They identified ZmNECAP1, which is also a transcriptional activator, as an interaction partner of Sts2 via Co-IP/MS. However, the authors did not investigate how ZmNECAP1 functions in leaf tumor formation together with Sts2. Therefore, the biological significance of the Sts2-ZmNECAP1 interaction remains unclear. In addition, NECAP1 has been shown to be involved in clathrin-mediated endocytosis in mammals (Ritter et al., PLoS Biology, 2013, PMID: 24130457). Furthermore, researchers recently published the involvement of clathrin-mediated endocytosis for effector translocation by fungal and oomycete pathogens in plants (doi/10.1093/plcell/koad094; doi.org/10.1093/plcell/koad069). Could NECAP1 enhance Sts2 translocation? Could it be possible that interacting with ZmNECAP1 confers enhanced activator activity

of Sts2? Additionally, the DNA binding activity of Sts2 is not tested, which could not be completely excluded. I also wonder whether ZmNECAP1 is able to bind DNA, which may facilitate Sts2 in executing its activator activity. In any case, the authors are encouraged to study how ZmNECAP1 functions together with Sts2.

2. In a previous publication of the same group, Sts2 of *U. maydis* was speculated to be under neofunctionalization to acquire the tumor-inducing function as its orthologs from another smut fungus *Sporisorium reilianum* do not complement the *sts2* mutant of *U. maydis*. In this study, the authors found that the transactivation domain (TAD) of Sts2 that is essential for its transcriptional activator activity is not conserved in *S. reilianum* orthologs. However, evidence of whether the *S. reilianum* orthologs exhibit activator activity is missing. It might also be better to perform a wider comparison of Sts2 orthologs in smut fungi (if found) and find the presence or absence of a functional TAD. This will help the authors to make a stronger claim that Sts2 of *U. maydis* undergoes neofunctionalization for tumor elicitation. However, it should be noted that there are other smut fungi which produce swelling structures, e.g., *Ustilago esculenta*. In addition, it would be interesting to directly test if the expression of *U. maydis* Sts2 in maize leaves by *S. reilianum* (or by bombardment) can induce the transcriptional reprogramming for hyperplasia.

3. It appears to be solid that Sts2 could induce certain gene expression of maize leaf development to form tumor structures, such as ZmGRF3, ZmGIF1, ZmYAB1, ZmWox5b. However, it remains unclear whether/how Sts2 activates the expression of those genes and which region(s) in the maize genome Sts2 is exactly associated with. To this end, the authors can perform CHIP-seq or CHIP-PCR to get more evidence.

Minor comments:

4. Line 32 and elsewhere: “Contrary, ...” to “On the contrary, ...”

5. Line 52 & 54: It should be “hyperplastic” and “hypertrophic”.

6. Line 55: Since See1 promotes tumor formation by re-activating plant DNA synthesis, the statement “...are still unknown” here sounds a bit strong. Please rephrase this point.

7. Line 69: It seems arbitrary to claim *Sporisorium reilianum* as “the closest smut relative” of *U. maydis*. *Pseudozyma hubeiensis* is probably the closest relative known to date.

8. Line 79: Please tone down the conclusion as it is too early to state that “Sts2 transcriptional activation is crucial to maintain the meristem like cell divisions...” according to the presented data.

9. Figure 1a: Could the authors show pictures of infected leaves and WGA/PI staining at 6 dpi when tumors begin to become pronounced? This will help to reach the conclusion that Sts2 is not involved in immune suppression at the early stage of infection.

10. Line 113 and elsewhere: It should be “mCherry”.

11. Figure 2e: The legend says that samples were collected at 3 dpi, but the text (Line 124) says 4 dpi. Please double-check it.
12. Line 139: It would be better to provide a multiple sequence alignment of Sts2 and its orthologs.
13. Line 149 and Figure 3f: Is DB-Sts2 Δ SP a typo (which is BD-Sts2 in Figure 3e)?
14. Figure 4a: An anti-Myc Western blot after IP should be provided.
15. Figure 4c: Precisely, the subcellular colocalization of Sts2 and ZmNECAP1 is in the nucleoplasm but not the nucleolus.

Reviewer #2 (Remarks to the Author):

A transcriptional activator effector of *Ustilago maydis* regulates hyperplasia in maize during pathogen-induced tumour formation

Weiliang Zuo et al.

The manuscript presents evidence that the corn smut fungus *Ustilago maydis* produces a secreted effector protein, Sts2 that acts as a transcriptional activator which promotes the division of tumour cells in maize, characteristic of the disease. They suggest that Sts2 interacts with a maize transcriptional regulator ZmNECAP1 and this leads to expression of a set of genes involved in cell division and development. This is an interesting paper and an example of a previously undescribed effector function for this fungal pathogen. The approaches taken are appropriate and normally present at least two independent lines of evidence for most of the conclusions made. There are some potential areas where improvement to the clarity of the study could be made and potential additional control experiments. Some other lines of enquiry are also suggested. I enjoyed reading this submission and there is considerable merit in the study.

1. One of the questions I often ask about *Ustilago maydis* papers of this kind is why the solopathogenic strain SG200 is exclusively used? I understand why this strain was originally made and why it makes *Ustilago* studies easier to carry out, but I do think the authors (and indeed most of the *Ustilago* community) should admit that it is not a natural strain at all. I do think that when important new effector functions like this are identified and are considered to be virulence factors, then the real proof should always be to generate a homozygous dikaryotic strain by deleting the gene in two compatible (e.g. a1 b1 x a2 b2) haploid strains. I am happy that subsequent detailed analysis can utilise SG200, as long as the authors are open about it being a proxy for the true pathogenic form of the fungus, but there should always be an experiment where the homozygous dikaryon is made and tested for virulence. I

have asked this of most major *Ustilago* papers that I have reviewed and seldom have the authors taken notice, but I am going to keep asking, because you all know you should really do this!

2. In Figure 1 I think the statistical tests showing that there are no significant differences in Figure 1c should be shown, or this may not be apparent. The axis legends are too small to read too. I think that Figure 1d needs to be reproduced much larger with a zoomed in inset of the hyperplasia tumour cells. This is a critical observation and I struggle to see the differences from the figure provided. I think this is very important and is the most significant part of Figure 1. I expect the images are of sufficient quality, but the size and magnification is inadequate.

3. I do not understand why the localisation data was carried out with the Pit2 promoter rather than the native Sts2 promoter? What was it necessary to over-express the effector to see it's localisation and how can you be sure this is not artefactual?

4. In Figure 2 I would prefer to see increased magnification of the localisations to match the line scan images, rather than large areas of black box. I can see that it would be good to show a wider field of view if a bright-field image was also shown, but as it isn't, I don't understand why these images were reproduced in this way. The Western blot in Figure 2e has no loading controls and cannot be shown like this.

5. Figure 3g is called out ahead of most of Figure 2, which is confusing, and should really be referring to Figure 3h, which is then in the wrong Figure?

6. In Figure 3a, the evidence that Sts2 is causing transactivation really needs a control. The best would be a *Ustilago* effector that doesn't show the same activity.

7. I really like the oci-infiltration transactivation assay in *N. benthamiana*. IT is a very neat idea and adds another layer of evidence that Sts2 acts as a transcriptionally activator. But I think the whole leaves should be shown, or a bright field image shown alongside.

8. In Figure 3h, it is hard to interpret these bar diagrams. I can see that there are more small tumours compared to large tumours in the Sts2 mutant, and the Tukey test supports this and the lack of complementation with the dTAD and Srad strains, but it isn't very informative for anyone outside of the *Ustilago* field. The untrained eye would conclude that every plant is displaying a lot of disease symptoms and Sts2 mutants cause slightly fewer than wild type. Is there a more informative way of showing this information, or some representative plants in the supplemental information?

9. The co-immunoprecipitation assay shown in Figure 4a is not very convincing. I would prefer to see some replicates or a better example of the actual IP western. Figure 4 c would be better with zoomed in images and a separate nuclear marker, such as H1-CFP or similar, so there is separate evidence that this is actually the nucleus.

10. Figure 5f needs better image, as other co-infiltration experiments earlier

11. There is no legend to Figure 5i. I also have the same comments about figure 5h and i. These are really difficult for non-*Ustilago* scientists to interpret. I am not convinced that we work out the severity or otherwise of disease symptoms in this way. They obscure rather than reveal differences in my view.

12. The text is too small throughout Figure 5.

In summary, there are some very good things about this study. The use of co-infiltration assays and the clever SRDX experiments, collectively provide evidence that Sts2 does act as a transcriptional regulator. Others have been found (e.g. Kim et al. 2020, as cited by the authors), but the evidence for action as a transcriptional activator is stronger in this submission than any previous study. My only concerns are that we do not really know the mechanism and function of ZmNECAP1. In fact there is very little information provided about this in the paper. I would have liked to see analysis of mutant and over-expression maize lines of ZmNECAP1, for example, as these would be important for defining its function as a potential co-activator for example. Notwithstanding that criticism (and it would be a lot of work to complete, so I wouldn't insist on it), I think the authors could provide a little more discussion of its potential role.

My other concerns, listed above are all about data presentation, clarity and controls. I am sure the authors can respond to these and improve the quality of some of the Figures. I am broadly confident in their conclusions and they are important and interesting.

Reviewer #3 (Remarks to the Author):

In this work, the authors identified and characterized a *U. maydis* protein Sts2, which functions as a transcriptional activator and induces the tumor cells formation in Maize. They observed that Sts2 can translocate into the maize cell nucleus to modulate virulence function. The authors further identified a novel plant transcriptional activator, ZmNECAP1, which interacts with Sts2. Their transcriptomics data revealed that Sts2 regulates many genes that are involved in leaf developmental processes, and those genes may regulate tumorigenesis in maize cells. Overall, the authors identified new regulators in *U. maydis* that induces tumor formation, which would benefit the understanding of the mechanisms of plant tumorigenesis. However, the authors can elaborate more about the biological functions of Sts2 and ZmNECAP1 in this work.

My comments are listed below:

Figure 3h and 5g. What are the letters "a" and "b" mean? Please indicate in the legend.

Figure 4a. It is suggested to have a figure (e.g. a Volcano plot) to show the quantitative MS co-IP data and to indicate the enriched ZmNECAP1 in *sts2* replicates. Given that Sts2 can translocate into nucleus, did the authors also identify any maize nuclear proteins from MS co-IP data?

Supplementary Table 1. Which enriched proteins in the table is ZmNECAP1? None of the Protein ID shows ZM00001eb221890. In addition, five out of 7 enriched proteins only identified one unique peptide. Having only one unique peptide does not provide enough data for quantitative analysis.

Figure 5a. Did ZmNECAP1 gene increase the expression level in SG200 or CR-Sts2 samples?

Figure 5I. I did not find the legend.

REVIEWER COMMENTS

Reviewer #1 (Remarks to the Author):

This manuscript by Zuo et al. presents that the *Ustilago maydis* effector Sts2 may act as a transcriptional activator to upregulate the expression of genes involved in maize leaf development, which leads to leaf tumor formation. As how *U. maydis*, a devastating maize pathogen, causes tumor formation is unknown, this study can have a broad significance.

The authors employ a multitude of approaches, such as cytohistology, comparative transcriptomics, transcriptional activity assay, and trans-repression assay, to reveal the functional mechanism of Sts2. Firstly, histological results indicated that Sts2 possibly promotes hyperplasia of bundle sheath cells in the leaf tissue without immune suppression. Secondly, they discovered that Sts2 is localized to the plant nucleus and exhibits transcriptional activator activity only when fused to a DNA-binding domain. A putative transcription factor of maize, ZmNECAP1, was identified as an interaction partner of Sts2. Lastly, the authors showed that the transcriptional activation activity of Sts2 is required to induce the expression of several maize genes involved in leaf development, which may explain leaf tumor formation.

Overall, this study is intriguing and reveals for the first time the transcriptional activator activity of *U. maydis* effectors, which extends our knowledge of the virulence mechanism of this model smut fungus. The characterized effector Sts2 is distinct from known *U. maydis* effectors as it exploits its transcriptional activator activity to upregulate regulators of maize leaf development for tumor formation without inhibiting plant immunity. The manuscript is well-written, and the data shown are generally solid. Nevertheless, the work has weakness in its current form, especially in the mechanistic aspect. Below are some comments for the authors to improve their manuscript:

Major comments:

1. The authors stated that Sts2 possibly interacts with other proteins to execute its activator activity as it does not contain a predicted DNA binding domain. They identified ZmNECAP1, which is also a transcriptional activator, as an interaction partner of Sts2 via Co-IP/MS. However, the authors did not investigate how ZmNECAP1 functions in leaf tumor formation together with Sts2. Therefore, the biological significance of the Sts2-ZmNECAP1 interaction remains unclear. In addition, NECAP1 has been shown to be involved in clathrin-mediated endocytosis in mammals (Ritter et al., PLoS Biology, 2013, PMID: 24130457). Furthermore, researchers recently published the involvement of clathrin-mediated endocytosis for effector translocation by fungal and oomycete pathogens in plants (doi/10.1093/plcell/koad094; doi.org/10.1093/plcell/koad069). Could NECAP1 enhance Sts2 translocation?

Response: We appreciate this comment with regard to possible NECAP1 functions. Particularly the recent findings on clathrin-mediated endocytosis of effectors provide interesting hints that are precious for further studies of eventual functions of ZmNECAP1 during *Ustilago*-maize interaction. We performed a **new experiment** showing that ZmNECAP1 interacts with ZmAP2 β , suggesting a potential role in clathrin-mediated endocytosis (**Supplementary Figure 2e**). We therefore included a section on this in the discussion part of the revised ms. (**line 332ff**). More detailed elucidation of ZmNECAP1

function, which is entirely unstudied so far, will be subject of a whole new story and part of our future work.

Could it be possible that interacting with ZmNECAP1 confers enhanced activator activity of Sts2?

Response: That's an excellent point! To address this, we performed a **new experiment** in yeast and found that the interaction of NECAP1 and BD-Sts2 enhanced its transcriptional activation (by testing the activity of reporter gene β -galactosidase). We added this **new result** in **line194ff** and **new supplementary figure 2d**.

Additionally, the DNA binding activity of Sts2 is not tested, which could not be completely excluded. I also wonder whether ZmNECAP1 is able to bind DNA, which may facilitate Sts2 in executing its activator activity. In any case, the authors are encouraged to study how ZmNECAP1 functions together with Sts2.

Response: Being a transactivator, we did not primarily expect Sts2 to directly interact with DNA. Nevertheless, we deployed bioinformatic tools to identify the known or typical DNA binding domain from Sts2 or NECAP1. Unfortunately, we couldn't identify any of that. Of course, this could not exclude these two proteins has DNA binding ability via an unknown domain. To be more clear in this point, we modified our description to address we could not identify known DNA binding domain in the text. We do not have experimental evidence to show whether Sts2 or ZmNECAP1 can bind DNA or not, since to our knowledge there is no exquisite experiment to test the protein-DNA interaction without knowing which DNA sequence the protein bind. Having this said, we'd like to point out that exactly this is the main goal for our follow-up studies: the identification of DNA sequences and functional characterization of the maize TFs being activated by the Sts2 effector, as well as the eventual functions of ZmNECAP1 (on which we can only speculate right now). This is now explained more clearly in the discussion part.

2. In a previous publication of the same group, Sts2 of *U. maydis* was speculated to be under neofunctionalization to acquire the tumor-inducing function as its orthologs from another smut fungus *Sporisorium reilianum* do not complement the *sts2* mutant of *U. maydis*. In this study, the authors found that the transactivation domain (TAD) of Sts2 that is essential for its transcriptional activator activity is not conserved in *S. reilianum* orthologs. However, evidence of whether the *S. reilianum* orthologs exhibit activator activity is missing. It might also be better to perform a wider comparison of Sts2 orthologs in smut fungi (if found) and find the presence or absence of a functional TAD. This will help the authors to make a stronger claim that Sts2 of *U. maydis* undergoes neofunctionalization for tumor elicitation. However, it should be noted that there are other smut fungi which produce swelling structures, e.g., *Ustilago esculenta*.

Response: We thank the reviewer for this suggestion. We added a **new result** showing the transcriptional activity of Sts2 orthologs from *S. reilianum* in **new supplementary figure 6b**. We also conducted a blast search and identified Sts2 orthologs in several smut fungi but not in *Ustilago esculenta*. We showed the alignment of these orthologs and the predicted TAD from these orthologs, see **line 298ff** and the **new supplementary figure 6c, d**.

In addition, it would be interesting to directly test if the expression of *U. maydis* Sts2 in maize leaves by *S. reilianum* (or by bombardment) can induce the transcriptional reprogramming for hyperplasia.

Response: We actually had the exact same idea and therefore we made effort to overexpress Sts2 in *S. reilianum*. However, Sts2 alone didn't induced any tumor formation when

expressed in *S. reilianum* (**Supplementary Figure 6a**), indicating that multiple factors (effectors) are required to activate this complex cellular process. We did not try to use bombardment assay because this would only overexpress genes in the epidermal cells while the bundle sheath cells are the predominant origin of tumor cells in leaves. We further discuss the new result in discussion part, see **line 349ff**.

3. It appears to be solid that Sts2 could induce certain gene expression of maize leaf development to form tumor structures, such as ZmGRF3, ZmGIF1, ZmYAB1, ZmWox5b. However, it remains unclear whether/how Sts2 activates the expression of those genes and which region(s) in the maize genome Sts2 is exactly associated with. To this end, the authors can perform ChIP-seq or ChIP-PCR to get more evidence.

Response: We actually had been trying to conduct ChIP-seq to study which genes are directly associated with Sts2. For this, we overexpressed Sts2-3×HA in *U. maydis*, anticipating that it would be sufficiently delivered into host cell nuclei. However, after crosslinking and IP with anti-HA magnetic beads, we did not get sufficient DNA for downstream sequencing. In short, we hope that in future we will be able to generate transgenic maize lines overexpressing Sts2, which may increase our chances to conduct ChIP-seq successfully.

Minor comments:

4. Line 32 and elsewhere: “Contrary, ...” to “On the contrary, ...”

Response: We changed it throughout the manuscript.

5. Line 52 & 54: It should be “hyperplastic” and “hypertrophic”.

Response: Corrected.

6. Line 55: Since See1 promotes tumor formation by re-activating plant DNA synthesis, the statement “...are still unknown” here sounds a bit strong. Please rephrase this point.

Response: We changed the description.

7. Line 69: It seems arbitrary to claim *Sporisorium reilianum* as “the closest smut relative” of *U. maydis*. *Pseudozyma hubeiensis* is probably the closest relative known to date.

Response: yes, *Pseudozyma hubeiensis* is the closest one. We therefore specified the description saying that *Sporisorium reilianum* is the closest pathogenic relative of *U. maydis*. See **line 70**.

8. Line 79: Please tone down the conclusion as it is too early to state that “Sts2 transcriptional activation is crucial to maintain the meristem like cell divisions...” according to the presented data.

Response: Has been changed. See **line 71**.

9. Figure 1a: Could the authors show pictures of infected leaves and WGA/PI staining at 6 dpi when tumors begin to become pronounced? This will help to reach the conclusion that Sts2 is not involved in immune suppression at the early stage of infection.

Response. We included the WGA/PI staining of 6dpi samples. See figure 1b

10. Line 113 and elsewhere: It should be “mCherry”.

Response: We corrected it throughout the manuscript.

11. Figure 2e: The legend says that samples were collected at 3 dpi, but the text (Line 124) says 4 dpi. Please double-check it.

Response: Thank you for pointing out this mistake. It should be 4 dpi, we changed the legend accordingly.

12. Line 139: It would be better to provide a multiple sequence alignment of Sts2 and its orthologs.

Response: We did a new analysis, and added a phylogenetic tree and sequence alignment of Sts2 orthologs we identified in smut fungi. We also annotated the predicted TAD in these orthologs. See new Supplementary figure 6c, d.

13. Line 149 and Figure 3f: Is DB-Sts2 Δ SP a typo (which is BD-Sts2 in Figure 3e)?

Response: Yes, it was a typo, it should be BD. We corrected it throughout the manuscript.

14. Figure 4a: An anti-Myc Western blot after IP should be provided.

Response: We performed a new Co-IP and provide a new figure 4b.

15. Figure 4c: Precisely, the subcellular colocalization of Sts2 and ZmNECAP1 is in the nucleoplasm but not the nucleolus.

Response: We thank this reviewer for pointing this out. We observed the co-localization of Sts2 and ZmNECAP1 in both in nucleoplasm and nucleolus, although the signal is less strong, so we think it's better to use nucleus.

Reviewer #2 (Remarks to the Author):

A transcriptional activator effector of *Ustilago maydis* regulates hyperplasia in maize during pathogen-induced tumour formation

Weiliang Zuo et al.

The manuscript presents evidence that the corn smut fungus *Ustilago maydis* produces a secreted effector protein, Sts2 that acts as a transcriptional activator which promotes the division of tumour cells in maize, characteristic of the disease. They suggest that Sts2 interacts with a maize transcriptional regulator ZmNECAP1 and this leads to expression of a set of genes involved in cell division and development. This is an interesting paper and an example of a previously undescribed effector function for this fungal pathogen. The approaches taken are appropriate and normally present at least two independent lines of evidence for most of the conclusions made. There are some potential areas where improvement to the clarity of the study could be made and potential additional control experiments. Some other lines of enquiry are also suggested. I enjoyed reading this submission and there is considerable merit in the study.

1. One of the questions I often ask about *Ustilago maydis* papers of this kind is why the solopathogenic strain SG200 is exclusively used? I understand why this strain was originally made and why it makes *Ustilago* studies easier to carry out, but I do think the authors (and indeed most of the *Ustilago* community) should admit that it is not a natural strain at all. I do think that when important new effector functions like this are identified and are considered to be virulence factors, then the real proof should always be to generate a homozygous dikaryotic strain by deleting the gene in two compatible (e.g. a1 b1 x a2 b2) haploid strains. I am happy that subsequent detailed analysis can utilise SG200, as long as the authors are open about it being a proxy for the true pathogenic form of the fungus, but there should always be

an experiment where the homozygous dikaryon is made and tested for virulence. I have asked this of most major *Ustilago* papers that I have reviewed and seldom have the authors taken notice, but I am going to keep asking, because you all know you should really do this!

Response: Indeed, in the *Ustilago* community the SG200 strain is often used rather than wild type crossings. We are fully aware that this is a recombinant strain, hence throughout the manuscript we carefully addressed it as SG200 to make it clear to the reader. In our opinion, if an effector knockout reduced SG200 virulence and this can be fully restored by genetic complementation, this is direct evidence that demonstrates a contribution to virulence (in a certain genetic background, on a certain host genotype, under controlled experimental conditions).

Nevertheless, we do agree with the reviewer that it can have an additional value to also confirm the effector's virulence functions in wild type crossings. We therefore generated *Sts2* mutants in the compatible strains FB1 and FB2, and used them for infection assays. These actually gave similar results as in SG200, confirming the virulence function of *Sts2* in these wild type isolates. We added this **new data in supplementary Fig1b**.

2. In Figure 1 I think the statistical tests showing that there are no significant differences in Figure 1c should be shown, or this may not be apparent. The axis legends are too small to read too. I think that Figure 1d needs to be reproduced much larger with a zoomed in inset of the hyperplasia tumour cells. This is a critical observation and I struggle to see the differences from the figure provided. I think this is very important and is the most significant part of Figure 1. I expect the images are of sufficient quality, but the size and magnification is inadequate.

Response: We changed the figure and point that out there is no significant difference between SG200 and CR-*sts2* at each timepoint, see **figure 1c**.

We appreciate the suggestion for the figure 1D and have modified the figure accordingly and now provide magnified views of the vasculature. See **figure 1e**.

3. I do not understand why the localisation data was carried out with the *Pit2* promoter rather than the native *Sts2* promoter? What was it necessary to over-express the effector to see its localisation and how can you be sure this is not artefactual?

Response: The main purpose of this experiment is not to show the localization of the effector, but rather to provide evidence to show that *Sts2* is a secreted protein being reflected by the accumulation of the effector-mCherry signal around the hyphae during in-plant colonization, which is in contrast to the non-secreted mCherry control. For this purpose, strong promoters that are highly induced at the early infection timepoint (around 1 dpi when epidermal cells are colonized) are used to generate sufficient levels of fluorescent signal. To further investigate the localization of *Sts2*, we overexpressed *Sts2* Δ SP-GFP in *N. benthamiana*, and, moreover, detected *Sts2*-3HA driven by its native promoter in the cell fractionation from infected maize leaves.

4. In Figure 2 I would prefer to see increased magnification of the localisations to match the line scan images, rather than large areas of black box. I can see that it would be good to show a wider field of view if a bright-field image was also shown, but as it isn't, I don't understand why these images were reproduced in this way.

Response: We modified the figures accordingly and show an increased magnification.

The Western blot in Figure 2e has no loading controls and cannot be shown like this.

Response: We are sorry to cause a misunderstanding here. Loading controls are often used to prove each loaded sample have similar/same total protein amount by showing an intense band from the loading. However, this is not applicable for the western blot shown in figure 2e. Here, we added different extraction buffers sequentially to extract/ fractionate different cellular content. The initial starting input material is same, and all fractionations are different from each other regarding to how much and which proteins inside. For example, the rubisco protein is not present in the nuclear and nucleoplasm fractions. Hence, instead of showing the input, the more important control is to show that the subcellular fractions are pure and not being contaminated with each other. For this, we provided the anti-UGPase, and anti-H3 as cytoplasmic and nucleolus/chromatin control, respectively. Similar information and data presentation is also showed in several publications (Wang, Ruyi, et al. *Current Biology* 26.18 (2016): 2399-2411, Xu, Fang, et al. *Molecular plant* 7.12 (2014): 1801-1804 and our main method reference Liu, Chang, et al. *Developmental cell* 44.3 (2018): 348-361.)

5. Figure 3g is called out ahead of most of Figure 2, which is confusing, and should really be referring to Figure 3h, which is then in the wrong Figure?

Response: We are sorry for the mistake. To eliminate the confusion, we deleted the call out of figure 3 before.

6. In Figure 3a, the evidence that Sts2 is causing transactivation really needs a control. The best would be a *Ustilago* effector that doesn't show the same activity.

Response: We agree with this comment and added the *See1* effector as a control. *See1* is another effector involved in tumorigenesis and it is also translocated into the host nucleus, yet it doesn't have transcriptional activation activity. Result is shown in new **figure 3a and line 142**.

7. I really like the *oci*-infiltration transactivation assay in *N. benthamiana*. IT is a very neat idea and adds another layer of evidence that Sts2 acts as a transcriptionally activator. But I think the whole leaves should be shown, or a bright field image shown alongside.

Response: We thank the reviewer for the appreciation and we totally agree with the comment that it is better to show the whole leaves. We modified the figure and now show original photos i.e. the whole leave under the GFP channel.

8. In Figure 3h, it is hard to interpret these bar diagrams. I can see that there are more small tumours compared to large tumours in the Sts2 mutant, and the Tukey test supports this and the lack of complementation with the dTAD and Srad strains, but it isn't very informative for anyone outside of the *Ustilago* field. The untrained eye would conclude that every plant is displaying a lot of disease symptoms and Sts2 mutants cause slightly fewer than wild type. Is there a more informative way of showing this information, or some representative plants in the supplemental information?

Response: Many thanks for pointing this out. We made a new figure and put a typical photo for each disease symptoms next to the legend. We also provide all the original disease scoring data and calculation of disease index for readers to understand (**see source data**).

9. The co-immunoprecipitation assay shown in Figure 4a is not very convincing. I would prefer to see some replicates or a better example of the actual IP western.

Response: Thank you for the comment. We provided a new western blot for Co-IP (**new Figure 4b**). We also showed the full images of our western blot in the **extended figure**.

Figure 4 c would be better with zoomed in images and a separate nuclear marker, such as H1-CFP or similar, so there is separate evidence that this is actually the nucleus.

Response: We added a new microscope image with co-localization of NECAP1-mCherry, Sts2-GFP and a nuclear localized YFP NLS^{sv40}-YFP together in **Figure 4e**.

10. Figure 5f needs better image, as other co-infiltration experiments earlier

Response: We agree. Now the complete leaf is visible and image has better resolution. See **figure 5f**.

11. There is no legend to Figure 5i. I also have the same comments about figure 5h and i.

These are really difficult for non-Ustilago scientists to interpret. I am not convinced that we work out the severity or otherwise of disease symptoms in this way. They obscure rather than reveal differences in my view.

Response: We added the missing legend to figure 5i.

With regard to the scoring and presentation of Ustilago infection data, we need to state that the different and qualitatively distinct symptoms that are caused by *U. maydis* do require the quantitative assessment of these individual symptoms. Thus, the way infection data is presented in Figs 5h and 5i is the typical scoring that is commonly established in the Ustilago community. To make this data more easily accessible to a broader readership we added a representative photo showing each symptom category.

In addition, we also provide all the original disease scoring data and the calculation of a quantitative disease index (**source data**).

12. The text is too small throughout Figure 5.

Response: We increased the text size for fig5.

In summary, there are some very good things about this study. The use of co-infiltration assays and the clever SRDX experiments, collectively provide evidence that Sts2 does act as a transcriptional regulator. Others have been found (e.g. Kim et al. 2020, as cited by the authors), but the evidence for action as a transcriptional activator is stronger in this submission than any previous study. My only concerns are that we do not really know the mechanism and function of ZmNECAP1. In fact there is very little information provided about this in the paper. I would have liked to see analysis of mutant and over-expression maize lines of ZmNECAP1, for example, as these would be important for defining its function as a potential co-activator for example. Notwithstanding that criticism (and it would be a lot of work to complete, so I wouldn't insist on it), I think the authors could provide a little more discussion of its potential role.

Response: We thank this reviewer for the positive comments on our work.

With regard to ZmNECAP1 we realize that its presentation in the manuscript was not clear enough.

In a nutshell: we believe that the major finding (and novelty) of this work is that a fungal pathogen deploys an effector that functions as a transcriptional activator (which to our knowledge is a novel finding) and that this effector is required to induce tumorigenesis in the bundle sheath via the activation of central developmental regulators of the plant. We think that these findings represent an important contribution to effector biology and significantly enhance understanding of pathogen-triggered formation of plant tumors. All this would basically be plausible even without knowing about ZmNECAP1 – however, we actually did find this protein in an interaction screen. Therefore, we found it relevant and justified to include this finding into this manuscript, despite the obvious fact that right now we have not fully understood, which role this protein actually has for the biological activity of Sts2. Based

on the data shown in the revised ms (also thanks to the suggestions of Reviewer 1, see above) we speculate that ZmNECAP1 i) may enhance the uptake of Sts2 into the host cell through clathrin-mediated endocytosis, and/or ii) it might recruit/enhance the transactivator activity of Sts2 (see new SFig 2d, e, showing that interaction of NECAP1 and BD-Sts2 enhances transcriptional activation in a yeast assay).

We did our best to better clarify the potential roles of NECAP1 as a helper/co-target of Sts2 and at the same time make clear that further elucidation of ZmNECAP1 and its interaction with Sts2 will be subject of an independent, future work.

My other concerns, listed above are all about data presentation, clarity and controls. I am sure the authors can respond to these and improve the quality of some of the Figures. I am broadly confident in their conclusions and they are important and interesting.

Response: We thank this reviewer for all the constructive suggestions and comments.

Reviewer #3 (Remarks to the Author):

In this work, the authors identified and characterized a *U. maydis* protein Sts2, which functions as a transcriptional activator and induces the tumor cells formation in Maize. They observed that Sts2 can translocate into the maize cell nucleus to modulate virulence function. The authors further identified a novel plant transcriptional activator, ZmNECAP1, which interacts with Sts2. Their transcriptomics data revealed that Sts2 regulates many genes that are involved in leaf developmental processes, and those genes may regulate tumorigenesis in maize cells. Overall, the authors identified new regulators in *U. maydis* that induces tumor formation, which would benefit the understanding of the mechanisms of plant tumorigenesis. However, the authors can elaborate more about the biological functions of Sts2 and ZmNECAP1 in this work.

My comments are listed below:

Figure 3h and 5g. What are the letters “a” and “b” mean? Please indicate in the legend.

Response: We have changed the statistic test and now indicate the significance in the figure by add “*”, “***” for the p value cutoff to show the significant levels.

Figure 4a. It is suggested to have a figure (e.g. a Volcano plot) to show the quantitative MS co-IP data and to indicate the enriched ZmNECAP1 in sts2 replicates. Given that Sts2 can translocate into nucleus, did the authors also identify any maize nuclear proteins from MS co-IP data?

Response: We thank the reviewer for this suggestion. We re-analyzed the MS data via Blast against the new maize v5 annotation. By this we now identified 6 candidates, which were exclusively detected in Sts2-3HA samples. However, since these candidates are uniquely detected in one kind of samples (only in Sts2-3HA samples), we couldn't use a volcano plot to show them together with other non-significant proteins. Instead, we added a table in the figure (**new Figure 4a**). We only focus on the significantly different protein in our MS data. From these 6 proteins, Zm00001eb210740 and Zm00001eb095440 encodes a translation elongation/initiation factor and CCR4-NOT, respectively, are potentially localized in nuclear.

Supplementary Table 1. Which enriched proteins in the table is ZmNECAP1? None of the Protein ID shows ZM00001eb221890.

Response: We apologize for this mistake and thank the reviewer for making us aware. We have been using maize B73 v3 annotation for our MS data analysis (this was done before v5

had been published). We now re-analyzed our data against the v5 annotation to be consistent with our RNA-seq analysis (see point above). By this, we now only identified 6 enriched proteins only present in our Sts2-3HA samples.

In addition, five out of 7 enriched proteins only identified one unique peptide. Having only one unique peptide does not provide enough data for quantitative analysis.

Response: With current state-of-the-art high resolution mass spectrometers we can precisely identify the intact peptide in question and by fragmentation and detection of fragments on MS2 level, the peptide's sequence can be determined using state-of-the-art search algorithms. Calculations have shown that any peptide, generated by digestion with trypsin and longer than 7 amino acids can uniquely identify a protein. Therefore we can claim the identification of a protein with one peptide only.

Figure 5a. Did ZmNECAP1 gene increase the expression level in SG200 or CR-Sts2 samples?

Response: We performed a qPCR and found NECAP1 expression being increased upon *U. maydis* infection. However, there is no significant difference between SG200 and CR-Sts2 infections. We added the result in **figure 4d**.

Figure 5I. I did not find the legend.

Response: We are sorry for this mistake and have added the missing legend.

REVIEWERS' COMMENTS

Reviewer #1 (Remarks to the Author):

In this revised version, Zuo et al. have adequately addressed the reviewers' comments. There are several minor comments for the authors to further improve their manuscript.

- 1) Line 49 & 155: Please delete "respectively" in the end as it is not grammatically correct in the context.
- 2) Line 51: The characteristic symptom caused by *Ustilago maydis*-"tumor formation" should be added here.
- 3) Line 56: It reads better with "The mechanisms for host tumor formation...".
- 4) Line 75-76: Although cited in Line 280, Schilling et al. (2014) is better to appear here as the context for specific leaf tumor induction (not tassel) by *Sts2*.
- 5) Line 147: Typo-"Gla4".
- 6) Line 183: I am still puzzled by the subcellular localization of *ZmNECAP1*, which seems to be inconsistent in Fig. 4e (cytoplasm, nucleus) and Supplementary Fig. 2b (nuclear membrane). Please carefully describe its localization pattern.
- 7) Fig. 3i: I suggest the photos indicating different scoring categories to be moved to SI.
- 8) Fig. 3e: It should be "35S" promoter with capital "S".
- 9) Line 716-717: The scalebar in Fig. 2d (left) and Fig. 2e (left) should not be both 200 μm .
- 10) Legend of Supplementary Fig. 2b: The Ref for RNA-seq data of *U. maydis* infection should be Lanver et al., *Plant Cell* (2018). Please replace Ref. 11 in the list.

Reviewer #2 (Remarks to the Author):

I am grateful to the authors for their comprehensive revision of the manuscript and the constructive manner in which they have responded to my comments. The changes to the Figures have enhanced the quality of the study in my view and made their study more compelling. I support publication of this version of the manuscript. It is an important contribution which enhances our understanding of effector biology in plant pathogenic fungi.

Reviewer #3 (Remarks to the Author):

My concerns have been appropriately addressed.

Reviewer #1 (Remarks to the Author):

In this revised version, Zuo et al. have adequately addressed the reviewers' comments. There are several minor comments for the authors to further improve their manuscript.

1) Line 49 & 155: Please delete “respectively” in the end as it is not grammatically correct in the context.

Response: corrected.

2) Line 51: The characteristic symptom caused by *Ustilago maydis*-“tumor formation” should be added here.

Response: corrected.

3) Line 56: It reads better with “The mechanisms for host tumor formation...”.

Response: corrected.

4) Line 75-76: Although cited in Line 280, Schilling et al. (2014) is better to appear here as the context for specific leaf tumor induction (not tassel) by *Sts2*.

Response: corrected.

5) Line 147: Typo-“Gla4”.

Response: corrected.

6) Line 183: I am still puzzled by the subcellular localization of *ZmNECAP1*, which seems to be inconsistent in Fig. 4e (cytoplasm, nucleus) and Supplementary Fig. 2b (nuclear membrane). Please carefully describe its localization pattern.

Response: We are sorry to cause confuse here. Both Fig. 4e and supplementary Fig. 2b showed that *ZmNECAP1* is localized in cytoplasm, nucleus and nuclear membrane, where nucleus is the place where both *ZmNECAP1* and *Sts2* are co-localized. As shown below from Fig. 4e, we can detect *St2*-GFP, NLS-YFP and *ZmNECAP1*-mCherry signal in the nucleus in nucleus, however, *ZmNECAP1*-mCherry showed two small peaks outside nucleus (as shown below) which resulted from its nuclear membrane localization. We revised the sentence to make it more clear. **See line 210-211.**

7) Fig. 3i: I suggest the photos indicating different scoring categories to be moved to SI. Response: As suggested by other reviewers, the photos for the symptoms are necessarily for the readers who are not familiar with *U. maydis* scoring. We therefore decided to keep it in the fig 3i but not in the supplementary material, which we thought is more convenient for readers.

8) Fig. 3e: It should be “35S” promoter with capital “S”.

Response: corrected.

9) Line 716-717: The scalebar in Fig. 2d (left) and Fig. 2e (left) should not be both 200 μm .

Response: Thank you for pointing out the mistake. After checking the original image, we realized there was a mistake in the scale bar of the Fig. 2d (left), we corrected it. And we also added the dashed rectangle to Fig. 2d (right), which was missing before.

10) Legend of Supplementary Fig. 2b: The Ref for RNA-seq data of *U. maydis* infection should be Lanver et al., Plant Cell (2018). Please replace Ref. 11 in the list.

Response: we added the reference.